# Physico-Chemical Approaches to Investigate Surface Hydroxyls as Determinants of Molecular Initiating Events in Oxide Particle Toxicity

**DOI:** 10.3390/ijms241411482

**Published:** 2023-07-14

**Authors:** Cristina Pavan, Rosangela Santalucia, Guillermo Escolano-Casado, Piero Ugliengo, Lorenzo Mino, Francesco Turci

**Affiliations:** 1Department of Chemistry, University of Torino, Via Giuria 7, 10125 Torino, Italy; cristina.pavan@unito.it (C.P.); rosangela.santalucia@unito.it (R.S.); guillermo.escolanocasado@unito.it (G.E.-C.); piero.ugliengo@unito.it (P.U.); 2“G. Scansetti” Interdepartmental Centre for Studies on Asbestos and Other Toxic Particulates, University of Torino, 10125 Torino, Italy; 3Louvain Centre for Toxicology and Applied Pharmacology, Université Catholique de Louvain, 1200 Brussels, Belgium; 4Nanostructured Interfaces and Surfaces (NIS) Interdepartmental Centre, University of Torino, 10125 Torino, Italy

**Keywords:** silica, metal oxide nanoparticles, molecular interaction, surface chemistry, molecular recognition pattern, adsorption, nano-bio-inorganic interface, poorly soluble particles, new approach methodology (NAM)

## Abstract

The study of molecular recognition patterns is crucial for understanding the interactions between inorganic (nano)particles and biomolecules. In this review we focus on hydroxyls (OH) exposed at the surface of oxide particles (OxPs) which can play a key role in molecular initiating events leading to OxPs toxicity. We discuss here the main analytical methods available to characterize surface OH from a quantitative and qualitative point of view, covering thermogravimetry, titration, ζ potential measurements, and spectroscopic approaches (NMR, XPS). The importance of modelling techniques (MD, DFT) for an atomistic description of the interactions between membranes/proteins and OxPs surfaces is also discussed. From this background, we distilled a new approach methodology (NAM) based on the combination of IR spectroscopy and bioanalytical assays to investigate the molecular interactions of OxPs with biomolecules and membranes. This NAM has been already successfully applied to SiO_2_ particles to identify the OH patterns responsible for the OxPs’ toxicity and can be conceivably extended to other surface-hydroxylated oxides.

## 1. Introduction

Silanols (≡Si-OH) are the hydroxylated moieties exposed at the discontinuity between the silica (SiO_2_) surface and the outer molecular environment. Silanols are strongly held to be involved in a complex molecular interplay with biomolecules and govern the biological effect of silica (nano)particles [1,2,3]. The occurrence of silanols in a few specific structural arrangements was recently proved to determine the membranolytic and, in turn, inflammogenic activity of amorphous and crystalline silica particles [1]. In search for a unifying structural determinant for the molecular initiating event (MIE) in particle toxicology, here, we are proposing a general approach to describe the hydroxyl (OH) functionalities on oxide particles and nanoparticles (OxPs).

OxPs are currently used in industry or proposed as promising new materials for industrial applications, ranging from catalysis, electronics, automotive, functional finishing, as well as in the agri-food sector and biomedical field. For example, silica is used in chemical and electronic industries, building materials, foods, cosmetic products, and drug delivery systems [4]. Titanium dioxide (TiO_2_) and zinc oxide (ZnO) are frequently used as UV filters in cosmetics and sunscreens [5]. Aluminium and cobalt oxide (Al_2_O_3_ and Co_2_O_3_) are used in paints, coatings, textiles, catalysis, sensors, medical imaging, and energy storage devices, respectively [6]. Iron oxide nanoparticles, in particular γ-Fe_2_O_3_ and Fe_3_O_4_, are used in magnetic resonance imaging, drug delivery, thermal ablation therapy, energy storage, and food packaging [7]. 

Some tremendous advances in material science and manufacturing are based on innovative use of OxPs. However, the increasing production and use of OxPs result in occupational, environmental, or biomedical exposure scenarios that may generate adverse effects on humans and the living organisms in general. The main routes of human exposure to OxPs are inhalation, ingestion, skin contact, and injection. Inhalation of SiO_2_ (nano)particles, for instance, is known to induce pulmonary inflammation, fibrosis, and occupational tumour development, especially when workers are exposed to crystalline forms [8,9]. Toxic effects of TiO_2_ particles are reported on some types of plants and algae. Moreover, membrane damage to mammalian cells, hepatotoxicity, nephrotoxicity, and pulmonary toxicity have been reported in rodents exposed to TiO_2_ particles [10,11]. Aluminium, cobalt, and iron oxide particles may cause cell membrane damage, cell death, increase in oxidative stress, and genotoxic effects. Studies on ZnO and CuO particles also showed possible cytotoxicity, oxidative stress, and DNA damage [12,13]. Toxic effects of a large panel of OxPs were recently reviewed by Sengul and Asmatulu (2020) and El Yamani et al. (2022) Through cheminformatics modelling of in vivo, in vitro, and in chemico data, both works highlighted the multifactorial parameters that control OxPs’ toxicity, which can be modulated by size, shape, surface, composition, solubility, aggregation, and particle uptake. 

OxPs exist in a plethora of chemical, structural, dimensional, and morphological variants (Figure 1), which in turn differentiate them for their key toxicologically relevant parameters, such as surface reactivity, micromorphology, and biopersistence. Such intrinsic variability has posed a huge concern for the toxicological assessment, predictivity, and safe design of (nano)materials and hindered the definition of structure-activity relationships (SAR) to model the biological hazard [14,15]. Currently, the most robust paradigm in particle toxicology includes the simultaneous presence of three key properties that derive from a combination of bulk and surface characteristics. To elicit toxicity effects, the particle must show: (i) a proper dimensional range to interact with organs/tissues, and target cells or to translocate to secondary target organs/tissues; (ii) a high chemical stability that determines longer residence time in the organism (i.e., biopersistence); and (iii) a high surface reactivity which may either result from a high specific surface area and chemically active surface groups, or from the presence of highly reactive transition metal ions exposed at the surface. A recent review of the literature on the toxicity mechanisms of transition metal oxide nanoparticles evidenced that the alteration of cell viability mainly depends on surface characteristics, i.e., available binding sites and surface charge, and the leaching of toxic metals in solution [16].

While the dimensional parameter is often controlled by the production methodology, particle solubility is defined by the structural and chemical properties of each oxide. Different OxPs might exhibit different toxicity mechanisms because of their different solubility pathways. As higher particle solubility often determines a lower biopersistence, poorly soluble particles generally pose a higher risk to human health and the environment than soluble particles. Nonetheless, the latter may elicit toxicity effects mainly because of the release of transition metal ions [17]. Cu^2+^ and Zn^2+^ ions leached out from oxides such as CuO and ZnO are well-known inducers of oxidative stress and pulmonary inflammation [17]. Upon cellular internalization, Cu^2+^ is known to promote redox cycles, which in turn trigger the typical Fenton-like generation of highly reactive ^•^OH radical and disrupt cell redox homeostasis [18]. Even if Zn^2+^ cannot directly participate in redox reactions, these ions can interact with sulphur, forming the triad of Zn, ROS, and protein thiols, such as in the case of metallothionein [19]. Similarly to CuO, Fe_3_O_4_ and particles that are contaminated by ferrous ions (Fe^2+^) are known to catalyse the Fenton or Haber–Weiss reactions and generate ^•^OH radical in solution [20]. 

If the particle-induced ROS production overcomes the physiological mechanisms of cellular defence (e.g., the GSH/GSSG equilibrium in the pentose phosphate pathway), an imbalance between systemic ROS and the cellular repair system against oxidative damage is generated. In particle toxicology, the ability of OxPs to cause oxidative stress is often advanced to explain toxic effects of OxPs at cellular level. Oxidative stress can lead to membrane damage, cell death, inflammation, or selective gene expression, depending on the hierarchical level of oxidative stress [21]. In addition to redox-reactive metal ion leaching, particles and nanoparticles are able to induce oxidative stress because of some peculiar unsaturated surface bonds and surface energy state. UV-irradiated nano-TiO_2_ is reported to induce alteration and toxicity in zebrafish embryos and larvae with a mortality rate as high as 36% after 96 h. The study invoked oxidative stress as the causal mechanism of action that elicited a toxic response in zebrafish [22]. A paradigm based on the different band gap energies of a panel of metal oxide nanoparticles has been proposed as a predictive model for oxidative stress and acute lung inflammation [17]. 

Although the ability to generate ROS and oxidative stress constitutes one of the principal injury mechanisms through which some OxPs can induce adverse health effects, the intrinsic capacity of (nano)particles to induce oxidative stress is not always related to their toxic activity. Lu et al. [23] analysed a panel of 13 metal oxide nanoparticles, including SiO_2,_ TiO_2_, Al_2_O_3_, ZnO, CeO_2_, and Co_3_O_4_, and showed that the potential to generate free radicals in acellular tests is relatively effective in predicting cytotoxicity in vitro, but not effective in predicting inflammogenicity in vivo. Similarly, crystalline silica damages cell membranes and induces cytotoxicity, inflammation, fibrosis, and severe lung diseases [24]. However, the intrinsic potential of pure silica to generate free radicals in a biomimetic environment is very low, and mainly assigned, when recorded, to iron ion contamination [1,25]. It should be noted that many OxPs, including silica, exhibit variable toxicity effects that are not merely a function of their solubility or surface energy state. Hence, besides the capacity of OxPs to generate oxidative stress, additional surface features should be involved in defining the mechanisms of OxPs interactions with biological systems.

Emerging evidence was recently advanced about the key role of the topochemistry of surface OH moieties of OxPs as a relevant contributor to the detrimental interactions established with biomolecules and biomembranes [26,27,28,29,30]. Many oxides (e.g., SiO_2_, TiO_2_, Al_2_O_3_,) and innovative nanomaterials (e.g., graphene oxide and nanoclays) expose hydroxyl groups at their surface, which can arrange in different orientations and relative distances depending on the structural constrain of each material. Production methodologies and post-production treatments may further modify the surface chemistry and hence the surface topochemistry of OH groups. Hydroxyl groups confer hydrophilic properties to the particles which are employed in important technological applications such as separation by chromatography and wettability [31].

A superhydrophilic/superhydrophobic CuO surface was recently tailored by UV irradiation, taking advantage of the metal–oxygen (Cu–O) bond breaking and related change in surface chemistry [32]. Upon UV irradiation, electron–hole pairs are commonly generated on several oxides, depending on their specific band gap, resulting in the possible formation of surface oxygen vacancies. These defective surfaces may interact with water molecules to form highly hydroxylated surfaces with higher wettability. Surface tuning is largely used to improve biocompatibility of implantable materials. For instance, the surface oxidation of the TiO_2_ layer on a Ti-based biomaterial and the creation of active hydroxyl groups (Ti-OH) was recently exploited to improve cytocompatibility, cell proliferation, and hemocompatibility, with respect to the pristine alloy [33]. 

The potential of hydroxyl groups to establish strong chemical bonds with external (bio)molecules depends on both the material composition and the OH distribution/topography. The crystal composition and structure determine a difference in electronegativity between both the OH group and the element peculiar to each solid, as well as impose geometrical constrains to OH groups. Surface OH groups can be discriminated for their relative position at the surface which in turn determines their mutual interactions. This generates different surface OH patterns that may also vary for the very same compound, depending on the lattice structure, crystal polymorphism, and thermal, chemical, or mechanical process that could determine a redistribution of the OH surface configuration. For instance, porosity may affect the density and spatial organization of OH groups available at the particle–biomolecule interface, leading to changes in the bio-nano cumulative interactions. This has been reported for mesoporous silica particles that were less membranolytic towards red blood cells (RBC) than their dense counterpart, because of the dramatic reduction of the number of silanol groups accessible to the cell membrane [34,35]. 

Surface OH groups interact with each other or with adsorbed molecular species by hydrogen bonding, and dynamic proton exchange among these species is expected [36]. Metal-bound OH groups are amphoteric in nature. When these species are surrounded by an aqueous medium, an equilibrium between the protonated and deprotonated species is achieved readily [37]. The degree of OH protonation/deprotonation is a function of the OH pattern peculiar to each sample, as each OH population displays different acidity constants [38,39,40]. Hence, both their surface structural arrangement and orientation, and their degree of protonation will define the particle surface topochemistry and the adsorption properties of hydroxyl sites at the oxide surface.

The interaction of silica (nano)particles with biomolecules is a well-established paradigm for the key role played by surface hydroxyls in defining the biological activity of materials. Pioneer works suggested that silanols might determine the bioactivity of crystalline silica [41]. More recently, a peculiar ≡Si-OH family, namely the nearly free silanols (NFS), was shown to play a crucial role in the interaction between particles and molecules, especially in adsorption and toxicity processes. NFS are now held as the key molecular species that impart to respirable crystalline silica (RCS) the potency to induce cell membrane disruption, activate in vitro inflammatory responses, and trigger acute inflammation in the lungs of animals [1,42]. Mechanistically, NFS-rich but not NFS-poor silicas were proven to induce cytotoxicity in macrophages, leading to the secretion of alarmin IL-1α as one of the early inflammatory events triggered by silica particles [43]. A positive correlation between the presence of surface NFS and the membranolytic capacity of silica was proven for synthetic and natural quartz, amorphous silica [1,44], crystalline silica polymorphs [45], and crystalline nanosilica prepared by ball milling [46]. Overall, recent research evidenced that NFS are a solid candidate to explain the different biological effects of many types and structures of silica specimens. 

The peculiar geometry of these silanol patterns, which are located at the well-defined distance of 4–6 Å apart, maximise the interaction energy with phospholipids (PLs) that make up cell membranes by interacting in a bidentate fashion with the PL polar head. Consequently, NFS were observed to induce a greater membrane perturbation than other silanol patterns and this perturbation was linked to the inflammogenic activity of silica [26]. The key role of NFS was also recognized in the formation of peptides catalysed by interstellar silica-based dust particles in the context of prebiotic chemistry [3,47]. Intriguingly, a specific role of NFS has also been proposed as driving the interaction with biomolecules for silicates and other OxPs. For instance, some nanoclays (e.g., kaolinite-rich layered aluminosilicate) were recently showed to expose outer OH group with interaction energies similar to NFS. For these clays, the capacity to damage membranes was related to surface OH terminations, namely silanols and aluminols, that were exposed at the crystal lattice boundaries. The membrane damage observed for amorphous metakaolin was driven by a specific OH family with a surface OH profile that could be assigned to the NFS previously evidenced in quartz [27]. 

Identifying peculiar reactive OH groups could be mostly useful in defining the Adverse Outcome Pathway (AOP) of OxPs, specifically in establishing the Molecular Initiating Event (MIE), i.e., the interactions between particles and target biomolecules that trigger the early molecular events leading to the pathological outcome (Figure 2). Mechanistically, damage to cell membranes has been reported as one of the main MIEs that elicits cellular toxicity of many OxPs, including the release of inflammatory and fibrotic mediator and cell death. In addition to silica, cell membrane disruption has been reported for TiO_2_, Al_2_O_3_, graphene oxide (GO), and many other oxides. Thus, the definition of reactive (in particular, membranolytic) surface OH patterns, as shown for silica (Figure 2), may represent a new paradigm for establishing toxic, but also catalytic, or more generally, active surface sites that interact with external (bio)molecules. In particle toxicology, a unifying model to predict the molecular mechanisms that drive the interactions between particle and biomolecules would allow the proposal of a predictive paradigm for nanoparticle toxicity pathways of oxides that are characterized by hydroxylated surfaces. 

The assessment of hydroxyl groups and their energy of interaction can rely on a set of experimental and analytical techniques that could be implemented by in silico molecular modelling. Among others, infrared (IR) spectroscopy methods are widely used to qualitatively and quantitatively assess surface OH groups on powders. Special experimental set-ups, such as H–D isotopic exchange with D_2_O vapour, can be easily implemented to grasp structural information about the OH population of the very first atomic layer of the bio-inorganic interface. Additionally, thermogravimetric analysis (TGA), titration procedures, temperature programmed desorption (TPD) in combination with mass spectroscopy, ζ potential titration of surface, and other spectroscopic techniques (XPS, NMR) could be used ([31] and references herein).

In this review, we present a new approach methodology (NAM) based on IR spectroscopy for identifying the surface OH sites that are responsible for the early events taking place when a xenobiotic particle enters a living organism, i.e., the interactions with biomolecules. The large-scale application of this approach to investigate the hydroxylated surface of oxides will favour the discovery and the consolidation of MIE in (nano)particle AOPs. Additional analytical methods to characterize surface hydroxyls and bioanalytical approaches that probe surface reactivity will also be considered to complete the pool of surface techniques aimed at identifying OH groups and defining their interactions with biomolecules. 

In this regard, models of cell membranes (i.e., liposomes and red blood cells) and molecular targets of the particle interaction (i.e., phospholipids and proteins) will be used to probe the particle surface reactivity. Experimental interaction models can be validated at the molecular scale by computational approaches. Since many of these aspects have been already clarified for silica, most of the examples will be drawn from studies related to silica, but the possible extension of this NAM to other OxPs will also be discussed. Far from being a comprehensive description of all the currently reported molecular mechanisms in OxPs’ toxicity, this work is limited to illustrating the physico-chemical and bioanalytical approaches that can be used to reveal and describe the surface OH of OxPs, mostly in a toxicological perspective. Several other features have been proposed as being involved in the toxicity mechanism of OxPs, such as the intrinsic capacity to generate ROS, and they have been extensively reviewed elsewhere [48].

## 2. Analytical Methods to Characterize Surface Hydroxyls

This section provides a concise overview of the main analytical methods available to qualitatively and quantitatively investigate the surface hydroxyl groups on several oxides. On oxides such as TiO_2_, SiO_2_, and Al_2_O_3_, surface hydroxyls form readily on surface defects upon exposure to water vapour and they facilitate, via hydrogen bond formation, the growth of molecular water layers [49]. When investigating those hydrated systems, one of the major challenges resides in discriminating the signals associated with dissociated water (i.e., hydroxyl groups) from those due to the adsorbed molecular water [50].

A first and straightforward method to quantify surface hydroxyls is potentiometric acid–base titration. Indeed, metal-bound hydroxyl groups are amphoteric in nature and the equilibrium between the protonated and deprotonated species is readily achieved.
M-O^−^ + H_3_O^+^ ⇆ M-OH + H_2_O ⇆ M-OH_2_^+^ + OH^−^(1)

When hydroxyls react with water they behave as weak Brønsted acidic sites (Equation (1)) and a qualitative description of the oxide “average acidity” may be derived from the thermodynamics of the acid–base equilibrium. In particular, pH measurements performed during acid–base titration experiments may be used to evaluate the net proton consumption and obtain surface OH densities [51]. Figure 3 reports the general scheme for acid–base titration of oxide particles and some experimental data obtained for TiO_2_ [52]. After contacting the oxide with water, surface and aqueous protons are exchanged and a specific net surface charge and protonation state are achieved (Figure 3A, scheme W). For example, by suspending 30 mg of TiO_2_ in water, the pH decreases to ~5, showing that the oxide surface exhibits acidic hydroxyls that partially deprotonate. Before starting the titration, a suitable quantity of acid is added to adjust the pH to ~3 (scheme S in Figure 3A and point S in Figure 3B). In this way we ensure that the surface is fully protonated and that the initial number of solution and surface protons is known. Figure 3C reports the values for the total added protons in the system (blue data) and the measured protons in solution determined from the pH (green data) during the titration. The difference between the two values accounts for the net surface proton excess/deficit (red data). We can identify three regions, marked by the vertical dashed lines in Figure 3B,C. On the left (region I), the H_3_O_(aq)_^+^ ions which are present in excess in the solution are titrated with a base; in the central part (region II), a reaction between the base and the surface protonated basic hydroxyls (_b_M-OH_2_^+^) or surface acidic hydroxyls (_a_M-OH) occurs; on the right (region III), upon consumption of the reactive surface protons, further base additions only increase solution OH_(aq)_^−^. In region II, the surface state changes from a proton excess condition to a deficit: we can, thus, identify the isoelectric point (IEP, i.e., the point where the surface H^+^ excess/deficit equals zero) and the equivalence points for both _b_M-OH_2_^+^ and _a_M-OH. The titration process, which ends at point E, removes protons from the system according to their relative acidities, following the order H_3_O_(aq)_^+^ >> _b_M-OH_2_^+^ > _a_M-OH >> _b_M-OH (the last step is not achieved in the experiment shown in Figure 3). Other possible approaches to determine the IEP and the particle surface charges are based on electrokinetic (or zeta) potential measurements [53,54].

Another technique which can be applied to the quantitative determination of OH functionalities is thermogravimetric analysis (TGA). TGA analyses the sample mass change as a function of time/temperature, while the sample is heated at a constant rate. This approach allows the quantification of the surface functionalities which can be thermally desorbed [55]. To check if the observed weight loss is due to the desorption of water or of other surface moieties, the TGA instrument can be coupled to evolved gas analysers based on mass spectrometry or Fourier transform infrared spectroscopy [56]. This approach allows the quantitative assessment of desorbed water, but the discrimination between molecular water and OH groups is not trivial, since it relies only on the different desorption temperatures [57]. Figure 4A shows a typical TGA curve for a sample of TiO_2_ nanoparticles. An initial steep weight decrease signals the desorption of the multilayers of physisorbed molecular water [58]. Smaller weight losses at higher temperatures are due to the removal of water molecules directly interacting with the surface Ti^4+^ centres and, finally, to the desorption of OH groups on high index facets and low coordinated defective sites [57,59]. These data can be processed to obtain the percentage of hydroxylated surface sites as a function of temperature (Figure 4B). Unfortunately, in the intermediate temperature range (i.e., ~200–300 °C) both tightly bound water molecules and the less stable hydroxyls can be removed [57], making a precise assignment of the signals troublesome since it relies only on TGA data.

Finally, several spectroscopic techniques can be employed to investigate surface OH groups. Here we briefly cover X-ray photoelectron spectroscopy (XPS) and nuclear magnetic resonance (NMR) spectroscopy. In the next section infrared vibrational spectroscopy is thoroughly presented.

XPS measures the number of electrons emitted by the sample as a function of their kinetic energy upon absorption of X-ray photons [60]. It can be employed to determine the water and hydroxyl groups content quantitatively by considering the signals related to photoelectrons emitted from O 1s levels. A proper spectral deconvolution should be performed to quantify the O 1s components related to strongly adsorbed H_2_O, OH groups, and lattice oxygens, which show slightly different binding energies [61]. The possible presence of an outermost layer of organic carbon contamination, which could contain oxygenated moieties, should also be taken into account by considering the C 1s emission peak. This procedure has been successfully applied to thin films where the oxide thickness was precisely known [62].

NMR spectroscopy monitors the transitions between specific nuclear spin states when the sample is exposed to an external magnetic field. The signals provide information about the kind of atomic nuclei and their chemical environment [63]. ^1^H magic-angle spinning (MAS) NMR spectra are very useful for the identification and quantification of hydrogen-containing species at oxide surface, but they can be limited by the chemical shift anisotropy and the H-H dipolar coupling [63]. These problems have been recently overcome by the increase of the spinning frequency. This approach has, for instance, been applied to silica samples allowing several families of silanols experiencing different hydrogen bond interactions to be quantified. In particular, in precipitated silica nanoparticles, isolated silanols showed sharp components at 1.8 and 2.1 ppm, while hydrogen bonded OH showed increased chemical shifts of up to 4.5 ppm [64]. ^17^O MAS NMR, performed after enrichment procedures and employing H_2_^17^O, has also been shown to be a powerful tool to study hydrated surfaces, illustrating the complex nature of interactions between oxides and water [65].

## 3. Procedures for Identifying the Surface Hydroxyl Families by IR Spectroscopy

### 3.1. Water and Hydroxyl Vibrational Modes

One of the most useful techniques to investigate the functional groups at OxPs’ surfaces is Fourier transform infrared spectroscopy (FTIR), which analyses the interaction of mid-infrared radiation (4000–400 cm^−1^) with matter, leading to excitation of the system vibrational modes. However, only molecular vibrations that are associated with variations in the dipole moment are infrared active [66,67,68,69]. The vibrational frequency of a bond, and thus the position of its associated IR band, is characteristic of the specific species, and can be calculated according to the following equation:(2)$$\tilde{\nu }=\frac{1}{2\pi c}\sqrt{\frac{k}{\mu }}=\frac{1}{2\pi c}\sqrt{\frac{km_{1}m_{2}}{m_{1}{+m}_{2}}}$$
where $\tilde{\nu }$ is the vibrational frequency in wavenumbers (cm^−1^), c is the speed of light, *k* is the force constant of the bond, which depends on the bond strength, and μ is the reduced mass, which depends on the *m*_1_ and *m*_2_ masses of the two species involved in the bond [66,68].

Molecular water shows three vibrational modes (Figure 5A): (i) a symmetric stretching mode (ν_sym_OH); (ii) an antisymmetric stretching mode (ν_asym_OH); and (iii) a bending or deformation mode (δH_2_O). The right part of Figure 5A also reports the vibrational modes of the hydroxyls directly bonded to a surface metallic centre, i.e., two different stretching modes (νMO-H and νM-OH) and one bending mode (δM-O-H) [70]. Figure 5B presents the mid-IR spectra of three different OxPs (SiO_2_, TiO_2_, and Al_2_O_3_). Spectra were collected after a brief outgassing step was carried out at room temperature to partially remove the multilayers of physisorbed water at the materials’ surface (black curves). Specifically, the band located at 1630 cm^−1^ is due to the bending mode of molecular water (δH_2_O) adsorbed at the OxPs surfaces. In the case of SiO_2_, this band is overlapped with the overtones and combination modes of its fundamental bulk modes (Si-O-Si), which are located in the 2150–1300 cm^−1^ region. For all the OxPs, a broad band, spanning from 3800 to 2500 cm^−1^, is ascribed to the superimposition of several νOH stretching modes of strongly adsorbed water molecules or hydroxyl groups. The position of the νOH bands depends on the type of interaction in which the O−H oscillators are involved. In particular, the IR absorption progressively shifts to lower wavenumbers and becomes more intense and broader when the OH oscillator is involved in stronger hydrogen bond interactions. The intensity of these van der Waals’ interactions is modulated by the local density, chemical nature, and geometrical arrangement of the OH species. Therefore, the IR investigation of the νOH modes represents a very powerful approach to grasp key information on the topochemistry of a hydroxylated surface [69,71,72].

Further information on the properties of the different hydroxyl families can be acquired by complete removal of the surface adsorbed water molecules. This is conventionally achieved by simultaneously outgassing and heating the OxPs. The resulting spectra after the thermal treatment for SiO_2_, TiO_2_, and Al_2_O_3_ are presented in Figure 5B (red curves). Thermal treatments produce the disappearance of the broad band centred around 3500 cm^−1^, which is due to the physiosorbed water. Specifically, SiO_2_ heated at 450 °C showed the elimination of the adsorbed water and the partial condensation of the surface SiOH, through the elimination of a H_2_O molecule and the formation of a Si-O-Si bridge. Following this treatment, three different types of hydroxyl families were observed: (i) silanol groups strongly interacting by hydrogen bond (3100–3650 cm^−1^) which still produce a broad band; (ii) OH groups weakly interacting with vicinal moieties; and (iii) isolated silanols showing a narrow band at ca. 3750 cm^−1^ [73,74,75,76]. A similar behaviour was observed for TiO_2_ heated in vacuum at 600 °C. The molecular H_2_O is desorbed from the sample surfaces, as testified by the disappearance of the δH_2_O signal and by the decrease of the νOH band in the 3600–2500 cm^−1^ range. At the same time, four different bands, which correspond to terminal and/or bridging positions of the different OH families not perturbed by H-bonding (i.e., isolated TiOH), can be detected after heating. These species are located at different surface positions of TiO_2_ (different faces, reactive corners, defective borders, etc.) [30,58,77]. Thermal treatment Al_2_O_3_ highlights bands of isolated OH, which are not involved in H-bonding and are directly bonded to Al centres, either tetrahedrally or octahedrally coordinated. As described above, these isolated species absorb IR photons at high wavenumbers (3800–3730 cm^−1^). Bands at lower wavenumbers (3710–3590 cm^−1^) correspond to bridged OH or OH groups which are weakly involved in H-bonding [78,79,80].

**Figure 5 ijms-24-11482-f005:**
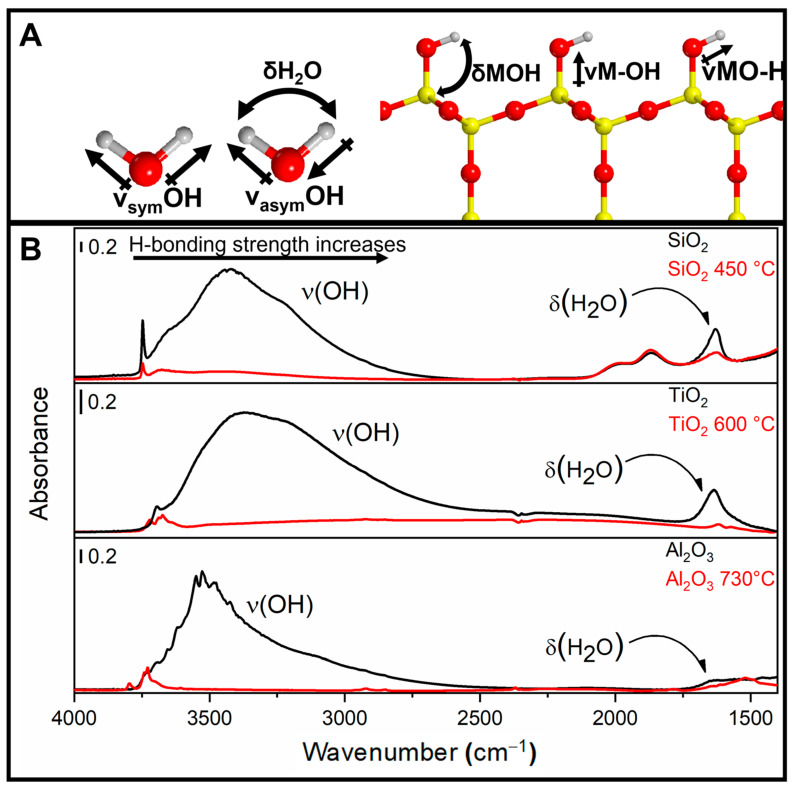
(**A**) Scheme of the main vibrational modes for H_2_O and OH groups (oxygen: red; hydrogen; grey; metal atom: yellow). (**B**) FTIR spectra of silica, titania, and alumina collected after short outgassing at room temperature (black curves) and after thermal treatments (red curves). Adapted with permission from [1,58].

### 3.2. H/D Isotopic Exchange Procedures

For some OxPs, the IR analysis of the surface hydroxyls could be complicated by the presence of structural OH families. A structural OH in fact shows IR bands that overlap with the bands of surface hydroxylated species that occur at the outermost atomic layer of an oxide particle. In this case, an isotopic exchange procedure allows the IR bands to be discriminated due to structural and superficial OH, as only the latter are accessible to deuterated water vapour (D_2_O, D = ^2^H) and are available for isotopic exchange. The methodology consists in performing an exchange of hydrogen present at the OxPs’ surface with deuterium (D), eventually transforming all the OH groups into OD groups. The use of isotopes in IR spectroscopy results in a shift of the absorption band positions. The shift is isotope-specific and can be predicted for H/D exchange by applying Equation (2): (3)$$\frac{{\tilde{\nu }}_{OH}}{{\tilde{\nu }}_{OD}}=\sqrt{\frac{\mu_{OD}}{\mu_{OH}}}$$
where the force constant is assumed to be equal for OH and OD bonds. This relationship allows the factor of isotopic shift OH/OD, being ca. 1.3736, to be obtained. However, the experimentally measured factors can differ significantly from the theoretical values and are generally lower. This deviation derives mainly from the fact that the anharmonicity of OD oscillators is significantly lower than the anharmonicity of OH oscillators. In addition, the anharmonicity of νOH vibrations is influenced by the formation of hydrogen bonds. On such a basis, the further decrease of the isotopic shift factor for hydroxyls is correlated with an increase of anharmonicity of the OH bond upon formation of the H-bond, which changes in a non-linear way as a function of the bond strength [81,82,83].

The recursive experimental procedure employed to obtain the isotopic exchange is summarized in Figure 6.

The initial outgassing of the sample is used to remove as many of the water molecules adsorbed on the material surfaces as possible. Figure 7A compares the spectrum of SiO_2_ before (red curve) and after (blue curve) outgassing. A remarkable reduction of the absorbance intensity in the OH stretching region (2700–3800 cm^−1^) and of the bending deformation mode of molecular H_2_O at 1630 cm^−1^ is observed due to the removal of physisorbed water on the OxPs surfaces. To completely exchange of surface OH with OD, several D_2_O vapour “admission—contact—outgas” cycles must be carried out. The cycles are repeated until a steady state of spectral invariance is achieved. Figure 7B compares the spectrum of an H/D exchanged silica sample before (red spectrum) and after (blue spectrum) outgassing. By further comparing the outgassed spectra of Figure 7A,B (blue lines), a strong decrease in intensity of the νOH band (3800–2500 cm^−1^) is accompanied by the appearance of a new broad band in the 2800–2200 cm^−1^ range, which corresponding to the OD stretching modes of the exchanged SiO−D. Significantly, the still-visible absorption of OH moieties signal the occurrence of hydroxyl species that could not be exchanged, likely because they are entrapped inside the particles and are therefore not accessible to the D_2_O molecules. In summary, H/D isotopic exchange allows the selective conversion of the surface OH groups into OD groups and to observe them in a spectral region without interferences from other signals, such as molecular H_2_O.

### 3.3. Sampling Techniques for IR Analysis

The analysis of surface OH groups on powders using IR spectroscopy can be conducted through various acquisition modes. The most conventional method is the transmission mode, where the IR beam directly passes through the sample. As the IR radiation interacts with the sample, it is in part absorbed, in part scattered, in part reflected, and in part transmitted. The transmitted IR radiation generates a spectrum that is typically represented as a plot of transmittance (the incident radiation that is transmitted through the sample) versus wavenumber (directly proportional to the energy of the radiation). The transmittance (*T*) is defined as the ratio of the transmitted intensity (*I)* over the incident intensity (*I*_0_) and takes values between 0 and 1:(4)$$T=\frac{I}{I_{0}}$$

However, transmittance is more commonly expressed as a percentage transmittance (*T*%) [68]. The transmission technique is limited by the preparation of the sample in a pellet form with an appropriate thickness. The thickness of the pellet is critical for accurate measurements as it influences the amount of IR radiation that is absorbed by the sample. In addition, to perform in situ experiments under a controlled atmosphere (i.e., for outgassing the sample and dosing H_2_O/D_2_O vapour pressure), the sample pellet can be held by a gold frame support and placed in a conventional IR cell, equipped with KBr (or CaF_2_) windows. The cell can be connected to a vacuum line and evacuated prior to establishing the desired atmosphere (Figure 8A).

Self-supported pellets of an appropriate thickness might not always be obtained due to a low specific surface or very high IR absorption of the sample. This scenario is very common for OxPs which often exhibit characteristics that are not compatible with transmission IR spectroscopy carried out on a self-supporting pellet. In those cases, transmission mode cannot be used, and the diffuse reflection mode is necessary. In diffuse reflectance Fourier transform infrared spectroscopy (DRIFT), the IR beam is directed onto the surface of the sample at an angle, and the reflected radiation is collected and analysed. The amount of IR radiation that is reflected depends on the chemical composition and properties of the sample. The interaction of the incident beam with the sample nanoparticles promotes different physical phenomena: reflection, diffraction, refraction, and absorption. These phenomena occur simultaneously and lead to the diffusion of the radiation over the entire area above the sample. Therefore, the interpretation of DRIFT spectra is less straightforward than transmission mode spectra. In particular, IR radiation can be specularly reflected from the sample surface (Fresnel reflection), it can undergo multiple reflections at the particle surface without penetrating into the sample and emerge from the surface at any angle relative to the incident beam (Fresnel scattering), or it can undergo the so-called Kubelka–Munk diffusion, which results from the penetration of the beam into one or more particles. This component of the radiation exits the sample surface at any angle, but because it has travelled through the particles, it contains information about the material absorption properties and, therefore, has the same informational content as a transmission spectrum. Clearly, diffuse reflection cannot be optically separated from specular reflection, but if the intensity of the latter is negligible compared to the former, the spectra are very similar to transmission spectra. The use of an experimental set-up with an appropriate geometry allows the minimisation of the undesired contributions caused by Fresnel reflection and scattering. 

The DRIFT cell includes a series of mirrors arranged in a convenient geometry (Figure 8B) that allows the IR beam to be focused onto the sample surface and maximises the amount of diffuse radiation that reaches the detector. One of the significant advantages of this technique is its ability to obtain infrared spectra of solid materials without requiring elaborate sample preparation. Indeed, the sample in the form of powder is directly placed in the sample holder. Additionally, for in situ measurements under a controlled atmosphere, an environmental chamber can be mounted on the sample holder. The atmosphere within the chamber can be controlled by connecting the DRIFT cell to a vacuum line which allows the vapour pressure of gas to be dosed or removed (i.e., H_2_O/D_2_O v.p. during H/D isotope exchange). In a DRIFT spectrum, the reflectance (*R*_∞_) is typically plotted as a function of the wavenumber (cm^−1^):(5)$$R_{\infty }=\frac{R_{\infty }\;sample}{R_{\infty }\;standard}$$
where the subscript ∞ indicates that the measurement is performed on a sample of “infinite thickness”, i.e., a sample that is thick enough (≥2–3 mm) to avoid both light transmission losses and contributions from rays reflected by the sample holder below the sample. The instrument measures relative reflectance, which is determined by calculating the ratio of the amount of radiation diffused by the sample to the amount of radiation diffused by the reference. A suitable reference is a non-absorbing material characterized by a high diffuse reflectance throughout the wavelength range of interest (e.g., KBr). Reflectance (*R*_∞_) is similar to transmittance (*T*) since the signal minima correspond to absorption maxima. This leads to spectra that closely resemble those obtained in transmission mode. 

However, the quality of a DRIFT spectrum can be influenced by several factors including refractive index of the sample, particle size, packing density, homogeneity, concentration, and absorption coefficient. One of the main causes of distortion in DRIFT spectra is the impossibility of optically separating diffuse reflection from specular reflection. Nevertheless, by taking the appropriate precautions it is possible to reduce undesired optical effects. For example, to minimise the effect of specular reflection, the sample particle size should be smaller than the wavelength of the incident radiation (generally between 780 nm and 1 mm). For highly absorbent samples, dilution in an IR non-absorbent matrix (KCl or KBr) allows for a deeper penetration of the incident radiation and minimises the contribution of specular reflection from the sample surface [67,84,85,86,87,88].

The fact that DRIFT measurements are highly dependent on the experimental conditions results in reflectance mode being less commonly used than the transmission mode. Additionally, the quantitative analysis of DRIFT spectra is more complex, as will be discussed in the next section. Despite these limitations, it is necessary to use reflectance mode in some cases. Moreover, the reflectance mode offers a noteworthy advantage as it is more sensitive to surface species than transmission measurements, thus improving the spectral signal–to–noise ratio in some cases. 

### 3.4. IR Data Treatment for Qualitative and Quantitative OH Assessment 

To assess the qualitative and quantitative aspects of hydroxyls on OxPs surface through the analysis of IR spectra obtained in transmission or reflectance mode, some data processing steps are necessary. In both instances, it is essential to convert the signal indicating the percentage of light passing through the sample (*T*%) or reflected from the sample surface (R%) into a signal that represents the amount of light absorbed by the sample. In the case of transmission mode, a logarithmic relationship exists between the absorbance (*A*) and transmittance values. As a result, an absorbance of 0 corresponds to 100% transmittance, while an absorbance of 1 corresponds to 10% transmittance.
(6)$$A=\log_{10}\frac{I_{0}}{I}={-\log}_{10}T$$

From a quantitative perspective, the well-known Lambert–Beer law describes a linear relationship between the absorption of light and the concentration of an absorbing substance in a sample. The law is expressed mathematically as:(7)A=εlc
where *A* is the absorbance, *ε* is the molar absorption coefficient, *c* is the concentration of the absorbing species, and *l* is the path length of the sample. The molar absorption coefficient, *ε*, is a measure of how strongly a molecule absorbs light at a particular wavelength being specific to the absorbing species [68].

Quantitative analysis becomes considerably more complicated in reflectance acquisition mode because the relationship between band intensity and concentration is no longer linear. The Lambert–Beer law is not directly applicable in this case due to the presence of scattering, which increases non-linearly as the wavelength of radiation decreases, making the optical path indeterminate. To make DRIFT spectra comparable to those obtained by absorbance measurements, it is necessary to process the data taking into account both absorption and diffusion. The most used mathematical model is the Kubelka–Munk function, *F*(*R*_∞_), which relates the diffuse reflectance *R*_∞_ with the absorption coefficient *K* and the scattering coefficient *S*:(8)$$F\left(R_{\infty }\right)=\frac{{\left(1-R_{\infty }\right)}^{2}}{{2R}_{\infty }}=\frac{K}{S}=\frac{2.303\varepsilon c}{S}$$

The behaviour of the sample depends on the ratio between *K* and *S*, rather than their absolute values. The term *K* includes the molar absorptivity coefficient *ε* (function of the wavelength *λ*), whereas the term *S* expresses the sample scattering ability (that is, an intrinsic property of the material that depends only on the particle size), but its dependence on *λ* is much lower than that of absorption. Therefore, assuming that the absorption coefficient *K* is sufficiently small (weak absorptions), and that the scattering coefficient *S* is constant over the entire energy range considered, the absorption spectrum acquired in diffuse reflectance (DR) and calculated with the K–M transformation allows a direct proportionality to be established between the intensity of the signals and the concentrations of the species that generate them. 

However, the K–M model is based on some theoretical assumptions and therefore has a limited range of applicability. Specifically, the relationship between the converted signal *F*(*R*_∞_) and concentration is only applicable if the sample is a weak absorber. In experimental practice, this means that only values of *F*(*R*_∞_) ≤ 1 (corresponding to sufficiently small absorptions) are linearly proportional to concentration. As previously reported, the underlying optical phenomena of DR spectroscopy are distinct from those of transmission measurements, but the information that can be derived from reflectance data is essentially the same, except for the relative signal intensity. In a DRIFT experiment, radiation undergoes paths of different lengths before emerging from the sample surface, and this affects the degree of light attenuation depending on the strength of absorption. In regions where the sample exhibits strong absorption, only the radiation that travels short paths is able to emerge from the surface, while for weak absorptions, even the light that travels longer paths is diffusely reflected, thus resulting in more intense signals compared to their transmission counterparts.

In quantitative evaluations, equivalent spectra represented in transmission or reflectance, and characterized by different values of *T* or *R*_∞_, produce different results when converted to absorbance *A* and K–M, respectively. The absolute values and areas of the signals in the K–M spectrum differ much more than one would expect from the values in reflectance. Signals with high *R*_∞_ appear less intense, while for low values of *R*_∞_ they appear more intense compared to an absorption spectrum (Figure 9A,B). This is a direct consequence of the type of mathematical equation applied during the conversion of transmission/reflectance spectra. By comparing the absorbance equation (*A*) with the K–M relationship (*f*(*R*_∞_)) as a function of *T* and *R*_∞_, respectively, it can be noted that the K–M function is almost flat for values of *R*_∞_ close to 1 and very steep for low values of *R*_∞_, while the absorbance curve shows less pronounced slopes (Figure 9C).

In addition, assuming that the accuracy of the measurements is not affected by instrumental errors but depends only on the difference in reflectance, it is possible to estimate the relative error associated with the Kubelka–Munk function by deriving the function *F*(*R*_∞_) with respect to *R*_∞_. From this analysis, Kortum [89] found that the range of reflectance values where the error on intensities is minimised in the application of the K–M law is between 0.2 < *R*_∞_ < 0.6. This means that the signal intensities in K–M are directly proportional to the number of oscillators when working within the 20–60% reflectance range. Outside of this range, the error increases rapidly. Despite its limitations and theoretical assumptions that restrict its measurements to mostly semi-quantitative analyses, DRIFT spectroscopy has a notable advantage over transmission spectroscopy in terms of sensitivity. In transmission spectroscopy, the signal–to–noise ratio (SNR) of a spectrum is directly proportional to the concentration of the sample, according to the Lambert–Beer law. However, in diffuse reflectance spectroscopy, as the concentration of the absorbing sample in the matrix decreases, *R*_∞_ tends towards unity and the SNR becomes proportional to the square root of the sample concentration. This feature is specific to DR measurements and gives DRIFT a considerable advantage over conventional sampling techniques for infrared microsampling [86]. Therefore, DRIFT spectroscopy is particularly useful for analysing samples that are challenging to prepare as self-supporting pellets, have granulometry that causes strong scattering (low specific surface area), and/or have a low concentration of surface species. These characteristics are commonly found in crystalline silica polymorphs, specifically quartz and cristobalite. The methodology to properly process IR data to obtain information on the OH population on these kinds of samples has recently been presented by our group [1,45,46].

The spectra needed for data processing are those related to the degassed sample before and after dosing the D_2_O vapour pressure (H/D isotopic exchange, see Section 3.2) collected in reflectance (or in transmittance) mode (Figure 10). After conversion to K–M (or absorbance), the two spectra (panels A’, B’) should be subtracted to isolate the contribution of the νOD modes, related to various surface silanol families.

As reported above, the interactions between silanols are the main factor that influences the position of the hydroxyls stretching signals (see Section 3.1). Thus, the closer the intersilanol distance, the greater the interaction force, and the lower the O-D stretching vibration frequency. In addition, the presence of H-bonds increases the signal intensity and broadens the νOD bands. Although the intensity of an IR band is usually proportional to the concentration of the generating species, the molar absorption coefficient ε of a surface silanol is strongly affected by inter-silanol interactions. The intensity of each component is not only proportional to the number of oscillators involved but also varies with adsorption frequency. A linear correlation between OH absorption intensity and wavenumber exists. A stronger H-bond results in elongation of the O-H bond and a higher dipole moment, leading to greater variation in dipole moment during vibration and a higher value of ε for strongly interacting hydroxyl species [82]. 

Carteret [90] has developed a model for scaling O-H stretching signal intensity based on the strength of the H-bond, thus making it independent from ε. The Carteret model shows the correlation between the integrated intensities of stretching SiO-H signals and wavenumbers. As the wavenumber decreases, leading to a stronger interaction between silanols, the integrated intensity of νOH signals increases significantly. The model provides a linear relationship that enables the extrapolation of the factors I(σ) for any value of wavenumber (σ). The scaling factor I(σ) calculated using the Carteret model allows for a correction of each νOH signal related to surface silanol species. By dividing the integrated area of each signal by the proper scaling factor, the signal intensity becomes independent of ε and a correct correlation between intensity and the number of oscillators involved in the signal can be obtained. The Carteret model was originally developed for the νOH pattern, and later adapted by some of us to the νOD pattern to quantitively analyse H/D exchange spectra. The lower anharmonicity of OD oscillators compared to OH oscillators, as well as the fact that the anharmonicity of OH/OD stretching vibrations is influenced by the formation of H-bonds, make it impossible to extend the Carteret model to the region of interest (2800–2200 cm^−1^) by applying the same shift factor for all identified components. Therefore, appropriate shift factors must be used to switch from one pattern to the other [83].

The process of adapting the Carteret model to the silanol νOD pattern involves the following steps:(i)fitting the original νOD profiles using components with a mixed Lorentzian–Gaussian character;(ii)shifting the frequency of each identified component in the νOD pattern to the νOH region using an appropriate isotopic shift factor obtained from experimental data taken from the literature [83];(iii)extrapolating the corresponding scaling factors I(σ) from the Carteret line based on the obtained νOH frequencies;(iv)dividing the integrated area of each component in the νOD pattern by the corresponding value of I(σ), to obtain the scaled components and recalculate the spectral profile.

A comparison between the original spectrum and the spectrum recalculated using the Carteret model shows a change in the relative intensities of the signals (Figure 11). Specifically, there is a decrease in the integrated areas of the signals at lower frequencies, which are attributed to strongly interacting silanols, and an increase in the bands at higher frequencies associated with weakly interacting silanols. This makes each band independent of the molar absorption coefficient ε, allowing the recalculated integrated area for each component to be effectively proportional to the number of oscillators that generate the signal itself.

In the study by Pavan et al. [1], the methodology proposed here has proven to be effective in identifying different silanol species. This approach allowed to establish a correlation between the membranolytic activity observed for some types of quartz samples and a peculiar silanol family, namely the NFS (see Section 4.1).

## 4. Bioanalytical Approaches to Probe Surface Hydroxyl Topochemistry

To probe differences in particle surface reactivity, including the topochemistry of surface OH groups, several tests that use model membranes or biomolecules are available. The bioanalytical tests discussed here, in addition to previous spectroscopic and analytical results, may be used collectively to unravel the molecular nature of the interaction of OxPs with biosystems. Both cell membranes and biomolecules, including phospholipids (PLs) and proteins, are important targets in the interaction of OxPs with living cells and such interaction may possibly trigger toxic outcomes [14,91,92,93].

### 4.1. Interaction with Cell Membrane Models (e.g., Red Blood Cells and Vesicles)

Direct contact of OxPs’ surfaces with cell membranes may compromise membrane integrity and provoke cytotoxicity. The use of red blood cells (RBCs) to test lytic effects on cellular membranes caused by OxPs’ surfaces has been one of the first assays used in particle toxicology for more than 60 years [94]. RBCs are non-internalizing cells; thus, the interaction with particles occurs only at the interface between the external surface of both the membrane and the inorganic solid. This simple yet informative test has been widely used for the assessment of haemocompatibility of NPs for drug delivery and imaging [95], and was also successfully exploited in inhalation toxicology. In this exposure route, RBCs play no part in the pathogenesis of lung diseases induced by particles. However, the haemolytic activity of several types of (nano)particles has been shown as the best predictor of their in vivo lung inflammation potential, with respect to other cellular tests that assess particle toxic effects on the respiratory system [23,91]. The membranolytic activity towards RBCs (i.e., haemolysis) may apply to OxPs whose mechanism of toxicity mainly involves particle surface features and interaction with cell membranes, as demonstrated in the case of SiO_2_.

In the case of different types of SiO_2_ particles, a positive correlation between haemolysis and the inflammasome-dependent release of inflammatory factors from alveolar macrophages (AM) has been shown (Figure 12A) [96]. The mechanistic reason resides in the lytic activity of SiO_2_ towards lysosomal vesicles, where particles are internalized after phagocytosis by AM. Lysosomal membrane perturbation leads to the activation of the inflammasome machinery that triggers the maturation and release of pro-inflammatory interleukins, including interleukin IL-1β and IL-18 (Figure 12B) [97,98]. Silica cytotoxicity and haemolysis correlated well with the available external surface area of the particles [99,100], suggesting that, for SiO_2_, the particle surface rules the interaction with biomembrane.

Furthermore, the haemolytic activity of SiO_2_ is reduced when OH groups are masked by polymers (e.g., poly(2-vinylpyridine-N-oxide) (PVPNO)) [41], proteins and lipids (e.g., albumin, lecithin, serum, plasma and corona) [101,102], and positively charged molecules, such as chloroquine [103] and aluminium compounds [41]. A decrease in haemolytic activity was also observed when replacing the OH groups with organosilane coatings on both crystalline and amorphous silica (nano)particles [34,104]. All these results supported the idea that the haemolytic activity of SiO_2_ particles depends on accessible silanol groups at their surface. 

In early studies, Pandurangi et al. [105] observed a variation of the haemolytic activity of SiO_2_ that was calcined at different temperatures. They observed that calcination at T below 500 °C increased the intensity of the silanol infrared band associated with some “free” silanols. The very same IR band was reduced when heating above 500 °C. Upon calcination, H-bonded silanols condense into siloxane bridges, and the SiO_2_ surface become progressively more hydrophobic [106]. By increasing the hydrophobicity of crystalline SiO_2_ with thermal treatments, the cytotoxic effect on macrophages, alveolar epithelial cells, and other cell models was reduced [107,108]. In more recent studies, it has been shown that calcination affects the NFS population which was related to the membranolytic potential of a large panel of crystalline and amorphous SiO_2_ samples [1,109]. By heating at intermediate temperature (i.e., 450 °C), the haemolytic activity of quartz increased and paralleled the increased population of NFS. When quartz was heated at 800 °C, the reduction of the membranolytic activity paralleled the reduction NFS, drawing a rather solid cause–effect between NFS and membranolytic activity (Figure 13A,B). Likewise, the increase and marked decrease of the membranolytic activity were correlated to the amount of surface NFS, modified by means of thermal treatments, and also on a pyrogenic amorphous nanosilica [1]. 

This observation was not limited to SiO_2_. By calcining the aluminosilicate kaolinite (i.e., a non-ion-exchanger layered silicate), a variation of its haemolytic activity was observed [110]. The NFS have also been identified on the dehydrated kaolinite (i.e., metakaolin) surface. NFS occurrence paralleled the variation of the metakaolin haemolytic activity well following heating at several temperatures (Figure 13C,D) [27]. Moreover, strong chemical treatments, such as etching with hydrofluoric acid (HF), caused a strong decrease of quartz haemolytic activity which was associated with a deletion of NFS (Figure 13E,F) [111]. Thus, the haemolysis assay allows the testing of modifications of the hydroxylated surface that could be induced by physical and chemical treatments severely affecting the surface.

**Figure 13 ijms-24-11482-f013:**
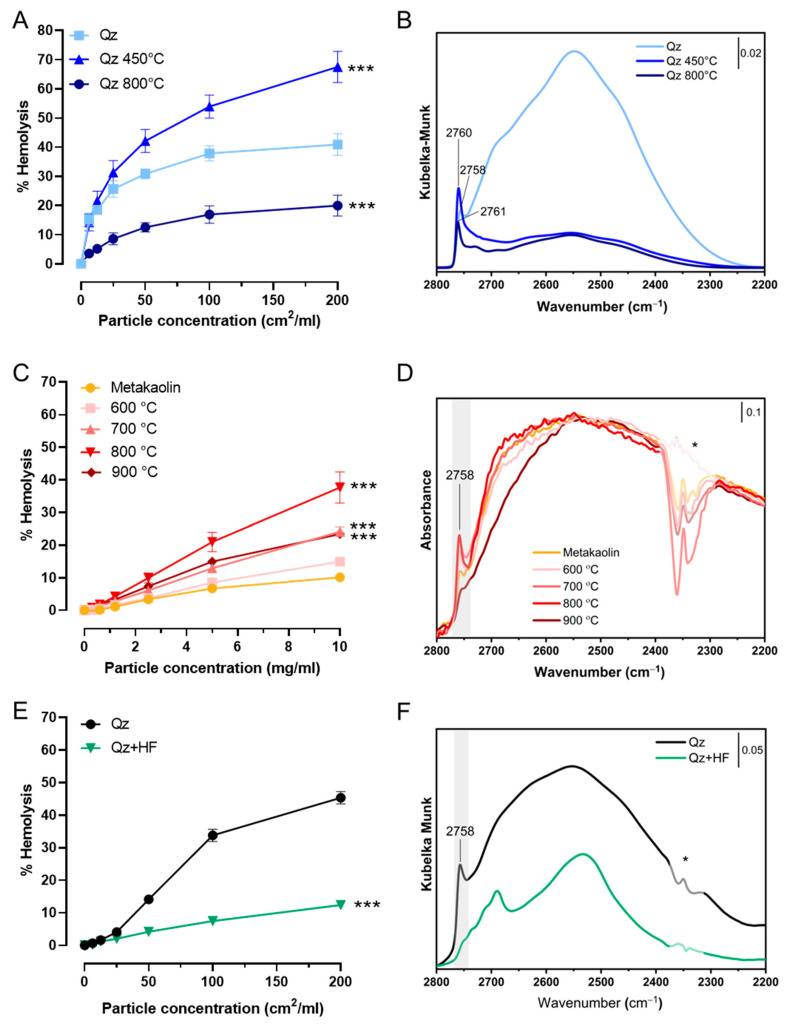
Modification of the hydroxylated surface by thermal and chemical treatments is related to particle membranolytic activity. (**A**,**B**) A pure quartz (Qz) calcined in air at 450 °C and 800 °C for 2 h; (**A**) Membranolytic activity (percent haemolysis) of pristine and calcined Qz samples, and Min-U-Sil 5 (positive reference particle); (**B**) Surface silanol distribution (after H/D exchange, reflectance IR spectra reported in Kubelka–Munk function) of pristine and calcined Qz samples. (**C**,**D**) A metakaolin (obtained by heating kaolin at 500 °C) calcined in air between 600 and 900 °C for 2 h; (**C**) Membranolytic activity (percent haemolysis) of pristine and calcined metakaolin samples; (**D**) Surface silanol distribution (after H/D exchange, absorbance IR spectra) of pristine and calcined metakaolin samples. (**E**,**F**) A quartz (Qz) etched with HF 0.1 M for 30 min; (**E**) Membranolytic activity (percent haemolysis) of pristine and etched Qz, and Min-U-Sil 5 (positive reference particle); (**F**) Surface silanol distribution (after H/D exchange, reflectance IR spectra reported in Kubelka–Munk function) of pristine and etched Qz. The peak at 2758 cm^−1^ is associated with NFS. * mis-compensation of atmospheric CO_2_. Data in (**A**,**C**,**E**) are mean ± SEM of three independent experiments; P values of treated samples compared to the pristine ones determined by two-way ANOVA followed by Dunnett’s post hoc test (mean effect): *** *p* < 0.001. Adapted with permission from [1,27,111].

A high haemolytic capacity has also been observed for micro- and nano-sized TiO_2_ particles [112]. The crystalline forms, in particular the anatase polymorph, resulted in much more membranolytic than amorphous TiO_2_, independently from the particle size range [112]. However, the mechanism of membrane perturbation has not been systematically investigated yet. A recent study [113] employed shape-engineered TiO_2_ particles [114] and proposed a correlation between the extension of the {001} facets of TiO_2_ nanocrystals and the haemolytic activity (Figure 14) and invoked the easier release of hydroxyl radicals from the {001} facet as the main mechanism of membrane damage.

Similarly to SiO_2_, the haemolytic activity of anatase TiO_2_ particles was inhibited by the addition of plasma proteins (i.e., albumin and γ-globulin) [112]. Considering that the RBC membrane is not sensitive to abiotic hydroxyl radical generation induced by particles [23,109,115], a possible involvement of specific population(s) of surface OH groups might be envisaged and would require further consideration. These aspects are particularly relevant for TiO_2_ toxicity and biocompatibility investigations, in which the well-known ROS-mediated photooxidative activity of UV-irradiated TiO_2_ should be disregarded [10,116]. In the absence of direct UV light, alternative surface-mediated pathways of TiO_2_ (nano)particle toxicity might be involved. TiO_2_ nanoparticles have been shown to induce cell death through lysosomal membrane permeabilization, cathepsin B release, and inflammasome activation [117]. This cascade has been proven to be dependent on several TiO_2_ physico-chemical characteristics, including size and crystal structure [118], but the surface features that are directly involved in the membrane permeabilization process have not yet been identified. 

Alumina (Al_2_O_3_) has also been shown to cause haemolysis in a particle size-dependent manner [119]. However, the doses used in the study by Vinardell et al. were very high and data were not normalized by specific surface area, which is essential when attempting to compare surface effects of particles largely differing in size, from micro- to nanometric. Lu et al. [23] showed that one sample out of four of Al_2_O_3_ nanoparticles was haemolytic and inflammatory but yielded a negligible generation of free radical species in solution. Once again, it is unlikely that the haemolytic and inflammatory effect is due to a direct particle-induced oxidative stressor. The observed haemolytic and inflammatory activity might rather be due to some activity of surface OH groups. Among the hydroxylated nanomaterials that are currently investigated as candidates for biomedical applications, it is also worth mentioning graphene oxide (GO). GO was found to be more membranolytic towards human erythrocytes than dehydroxylated graphene sheets [120]. Notably, GO has been shown to directly perturb the membrane lipids in primary human neutrophils [121]. However, little is known about possible relationships between haemolytic properties and OH distribution for these oxides. 

In addition to RBC, lipid mono- or bi-layers and vesicles made up of organized phospholipids (PLs) are used as simplified surrogates of cell membranes [92,93]. The use of these models allows the control of the physical and chemical parameters of the membrane, by tailoring the type and ratio of PLs in the bilayer. Several particle physico-chemical parameters, including size, shape, agglomeration/aggregation state, and differences in surface chemistry (e.g., surface hydrophilicity/hydrophobicity and charge), have been reported to influence the interaction with model membranes [122]. Several studies demonstrated that crystalline and amorphous SiO_2_ (nano)particles may destabilize zwitterionic vesicles of phosphatidylcholine (PC) lipids in different size ranges [26,100,123,124,125,126]. The same effects have been observed for colloidal and pyrogenic nanosilica interacting with mixed vesicles which mimic the PL composition of lung cells [100]. Silica nanoparticles ranging from 50 to 500 nm induced dose-dependent dye leakage from 1,2-dioleoyl-sn-glycero-3-phosphocholine (DOPC) liposomes [126], which is indicative of compromised membrane integrity, with similar efficiency when compared by surface area dose [100]. When the SiO_2_ surface was decorated with amino or carboxyl groups, as well as passivated with proteins or a PL bilayer, these coatings considerably reduced or prevented membrane destabilization [100,126]. Thus, as for RBC lysis, this has been assigned to a partial or total masking of the surface silanol groups. It has been demonstrated that occurrence of NFS on silica increases lipid order and membrane perturbation in zwitterionic PC liposomes [26]. 

Tuning the properties of model membranes, such as membrane surface charge by varying PL headgroups, cholesterol content, and size range, can add information to clarify the nature of the OxPs–membrane interactions. Inclusion of cholesterol in zwitterionic liposomes attenuated the increase in lipid order induced by SiO_2_ [29]. Negatively charged liposomes made up of mixed 1-palmitoyl-2-oleoyl-sn-glycero-3-phosphoglycerol (POPG)/DOPC reduced membrane destabilization of liposomes challenged with SiO_2_ [126]. Incorporation of increasing amounts of PG into zwitterionic PC vesicles progressively reduced membrane destabilization of liposomes that were exposed to SiO_2_ [100]. Liposomes of negatively charged 1,2-dioleoyl-sn-glycero-3-phospho-L-serine (DOPS) were not affected by NFS-rich silica [26]. It has also been reported that negatively charged silica and alumina-coated positively charged silica preferentially damage giant vesicles of opposite charge [125]. This indicates that long-range electrostatic forces may be essential for silica adhesion on membrane surface. The adhesion is reduced when bare SiO_2_, which is negatively charged at physiological pH due to deprotonated silanols, is incubated with anionic liposomes because of an increased electrostatic repulsion. However, even if important for adhesion, electrostatic force is not the main cause of membrane lysis. Membrane damage was not observed when vesicles were exposed to oppositely charged NPs [127]. A preferential affinity of silanol groups for the phosphocholine headgroup has been established and proposed as the cause of membrane destabilization [26,100,125,128,129]. The high affinity of silanols for PL phosphate groups, engaged in short range H-bonds, as well as the electrostatic, dispersion, and de-solvation contributions generated by the interaction between silanols and quaternary ammonium groups on PLs that expose PC or sphingomyelin (SM), might explain the strong affinity between several zwitterionic physiologically relevant PLs and silica. The key role of the quaternary ammonium groups in the interaction with SiO_2_ is also highlighted by the negligible binding and rupture of vesicles made up by PLs that expose a zwitterionic headgroup containing a primary amine (e.g., phosphoethanolamine PLs) [100]. At the cellular level, zwitterionic PC and SM are preferentially localized in the outer leaflet of the plasma membrane and on the luminal side of lysosomal membranes [130]. Nanoscale protrusion, also named “lipid rafts”, that are enriched in SM have been also observed on the surface of RBC [131]. This would explain why SiO_2_ induces a remarkable lysosomal perturbation and RBC/cell membrane permeability. The interaction between SiO_2_ and choline headgroup is thought to restrict lipid motion and cause local membrane gelation, reducing fluidity [125,132]. Stiffening of the membrane may disturb membrane signalling and trafficking, being a possible reason for cytotoxicity. 

A similar gelation mechanism has been described for other OxPs–membrane interactions. Dose-dependent leakage and lysis of lipid membranes has been demonstrated for TiO_2_ NPs that were put in contact with zwitterionic and negatively charged liposomes [133,134]. However, the molecular mechanisms that describe the interaction of TiO_2_ are still relatively unexplored and might differ from what is proposed for SiO_2_. In some cases, TiO_2_ nanoparticles did not exhibit the lytic activity towards PC liposomes [100], which is indicative of a certain variability of the interaction efficiency of TiO_2_ with membranes. This could be related to the different surface topochemistry of TiO_2_ hydroxylated/charged species and ad hoc investigations are required to understand the interaction mechanism at the molecular scale. Furthermore, different mechanism might be postulated for Fe_2_O_3_, which did not induce membrane destabilization of both DOPC and DOPS liposomes [134]. Studies using supported lipid bilayers and liposomes have demonstrated that GO interacts with and disrupts lipid membranes, but no mechanisms are currently being advanced [135,136].

### 4.2. Interaction with Biomacromolecules and Cell Membrane Components (e.g., Lipids, Amino Acids, Proteins)

When OxPs encounter a biological milieu/fluid, their surface is immediately covered by biomolecules (e.g., surfactants, amino acids, proteins). This biomolecular corona confers to the particles a new surface identity, which may determine cellular and tissue responses and (nano)particle fate in the body [137,138]. The biomolecule-solid surface interaction may either alter the particle surface properties or induce functional and structural changes to the biomolecules. It is well known that, besides particle size and nanomorphology, the material composition and surface chemistry determine the affinity and amount of biomolecules adsorbed on the surface of the (nano)particles and the extent of biomolecule alteration [139]. Surface charge, electron transfer capability, and functional groups play an important role in defining the affinity of proteins for the surface and the extent of surface-driven conformational changes. These features confer distinct proteomic/lipidomic fingerprints to the (nano)particles upon irreversible adsorption of biomolecules [137].

In general, OxPs with hydrophobic surfaces adsorb more proteins with a higher affinity and induce greater conformational alterations than hydrophilic surfaces [140]. In that sense, biomolecules can be used as a probe to check the surface properties of differently prepared (nano)oxides. In general, the hydrophilicity of an oxide surface is largely influenced by the type, amount, and geometrical arrangement of hydroxyl groups which are able to form H-bonds with water molecules [141]. For SiO_2_, hydrophobicity increases when silanols are condensed into siloxane groups by heating at temperatures above ca. 400 °C. Variation in the number and pattern of OH groups on SiO_2_ have been shown to affect/regulate the interaction with biomolecules. Even a slight variation in the silanol density on silica NP/flat surfaces was shown to alter the adsorption of fibrinogen at physiological concentrations, with the main consequences being for blood compatibility [142]. By increasing the hydrophilicity of SiO_2_, the amount of fibrinogen interacting with the surface decreased. The affinity of fibrinogen molecules for the SiO_2_ surface was, however, higher for hydrophilic surfaces, and this inhibited fibrinogen self-assembly. 

When inhaled into the alveolar space, particles interact with a continuous surfactant film, and may become coated with phospholipid bilayers. Zwitterionic self-assembled PC-exposing PLs, the main constituent of the lung lining layer (80%) [143], but not negatively charged self-assembled PLs (e.g., serine-rich PLs), have been shown to selectively interact with NFS on silica surface (Figure 15A). Zwitterionic PLs that expose a phosphocholine group (e.g., DOPC) hindered the NFS signal and the membranolytic activity of a NFS-rich amorphous silica. When the same silica was incubated with a negatively charged phosphoserine PL (e.g., DOPS) (Figure 15A,B), no inhibitory effects on NFS and membranolysis were observed. This further suggests a mechanism of molecular recognition between NFS and phosphocholine PLs, including SM and PC which make up cell membranes, and including RBC lipid protrusions (Figure 15C) [26]. 

The number and the distribution of surface OH was also related to the manner in which proteins are adsorbed on the surface of GO and Au NPs [144]. The adsorption of human serum albumin (HSA) and immunoglobulin E (IgE) for a set of finely tailored, hydroxylated (i.e., pure, partly, and fully hydroxylated) graphene and Au NPs was inversely correlated to the availability of OH groups on the surface, in line with what was observed for the adsorption of fibrinogen on silica [142]. The increased presence of hydroxyl groups also reduced the alterations to the secondary structure of HSA and IgE, minimising the possible denaturation effect. Surface-modified NPs interacted in a different manner with proteins. The TiO_2_ NPs that had more OH groups on their surface showed a stronger binding affinity to plant proteins [145]. Superparamagnetic Fe_3_O_4_ NPs (SPIONs) functionalized with amino- and carboxyl groups showed different corona compositions and cell toxicities. Some proteins (e.g., albumin) adsorbed only on positively charged amino-coatings and others (e.g., fibrinogen) only on negatively charged carboxyl-coatings [146].

## 5. Detailing the Molecular Interactions with Modelling

Vibrational spectroscopy represents a very powerful tool to identify and quantify the different hydroxyl families exposed at the OxPs’ surfaces. Nevertheless, the establishment of a precise correlation between a specific spectroscopic signal and the chemical nature as well as the local structural environment of the corresponding OH groups is complicated by the large number of different surface structures possibly exposed by (nano)particles. The interpretation of the spectroscopic data on OxPs’ samples can be assisted by comparison with surface science studies performed on well-defined single crystal surfaces [147,148]. However, this approach is not applicable to all systems and, moreover, single crystals do not expose all the defective and low coordinated sites present in (nano)particles. Therefore, the use of ab initio modelling can be crucial for a proper interpretation of the experimental results [149,150,151], allowing the elucidation of the OH structures and interactions at the molecular level. Different levels of theory, ranging from the more accurate post-Hartree–Fock and density functional theory (DFT) methods to approximated semiempirical and classical force field approaches (vide infra), can be adopted depending on the complexity of the oxide model and the amount of adsorbed water (from isolated OH to water multilayers) needing to be simulated [76].

For instance, Chillazet et al. [152] employed DFT GGA calculations to study the hydroxyl groups on MgO, γ-Al_2_O_3_, and anatase TiO_2_ at the molecular level. They investigated the impact of the main geometric (hydrogen bonding, coordination, surface symmetry) and electronic (nature of the metal–oxygen bond, M–O) parameters on the structure, stability, and vibrational properties of the surface hydroxyls. They also pointed out that the strength of the M–O bond influences the strength of non-dissociative water adsorption at the surface of different oxides. Conversely, water dissociation is favoured by the stronger basicity of O^2−^ anions in ionic oxides, such as MgO. The same groups of authors compared experimental IR spectra and periodic models of different anatase TiO_2_ surfaces to correlate the intensity of the specific IR bands with the OxPs morphology [153]. They also coupled ab initio calculations and a thermodynamic model to assess the surface hydration state under various temperature and water pressure conditions [154]. A similar approach was adopted by Viñes et al. [155], who performed a detailed hydroxyl identification combining DFT PBE calculations and IR spectroscopy. They showed that OH vibrations can be employed as fingerprints of ZnO surface morphology, being sensitive to specific exposed surfaces and to the water coverage.

A more complex case is represented by the modelling of amorphous OxPs where the conventional approach which relies on the use of experimental crystal structure data as a starting point for the simulation cannot be adopted. In this field, Ugliengo et al. [156] studied amorphous silica surfaces and built a derived model, starting from a bulk cristobalite with 192 atoms in the unit cell and then running molecular dynamics at a very high temperature, followed by cooling it down to room temperature, to obtain an amorphization of the starting material without introducing defects in the final structure. From this structure they obtained several surface models with different OH densities (Figure 16A) that were employed to simulate IR spectra (Figure 16B). To account for the nonuniform hydroxyl density of a real SiO_2_ surface, the calculated spectra from each of the considered models were convoluted, adopting suitable weights to match the expected experimental populations and obtaining a remarkable agreement with the experimental spectrum of an Aerosil 300 silica outgassed at 150 °C (which is expected to expose ca. 4 OH nm^−2^) (Figure 16C).

Recently, the use of realistic explicit oxide nanoparticle models, containing hundreds of atoms, allowed the limitations of the idealized infinite surface periodic models to be overcome, namely, those which do not provide a proper description of the undercoordinated sites located at the vertices, steps, and corners [157]. For instance, Mino et al. [30] employed a TiO_2_ nanoparticle model with 252 atoms and adsorbed on it up to 55 H_2_O molecules in a dissociative way. By simulating the vibrational frequencies at the DFT level and comparing them with experimental IR spectra (Figure 17), they were able to assign the different IR signals to hydroxyl groups located in different sites (i.e., nanoparticle apical, edge, and equatorial regions) and experiencing a different degree of H-bonding interactions.

Modelling can also play a crucial role in describing the interactions between OxPs surfaces and biomolecules/membranes. In this respect, understanding the structure and dynamics of proteins is essential for elucidating the molecular mechanisms of these processes. One of the most widely used computational methods for studying protein structure and motion is the molecular dynamics (MD) simulation [158]. MD simulation is a technique that, when used to describe very large systems such as proteins, uses classical mechanics to model the motion of atoms in a system over time. It requires a mathematical function and a set of parameters that describe the energy of the system as a function of the atomic coordinates. This function approximates the true quantum mechanical wavefunction, commonly referred to as a force field [159]. The force field consists of terms that represent different types of interactions between atoms, such as bonded (covalent) interactions, non-bonded (van der Waals, electrostatic, and hydrogen bonding) interactions, and solvation interactions. The parameters of the force field are derived from experimental or quantum mechanical data and reflect the geometric and energetic properties of interatomic interactions. The partial derivatives of the force field for the atomic coordinates yield forces that can be used to integrate Newton’s equations of motion and propagate the system through time using numerical algorithms. The most common force fields for proteins simulation are Amber [160], CHARMM [161], GROMOS [162], and OPLS-AA [163], each with its own strengths and weaknesses. Therefore, choosing an appropriate force field for a specific protein system and research question is an important step in MD simulation. Another very delicate step in the protein simulation is to properly set the protonation/de-protonation state of the various amino acids of the polypeptide chains. This is usually done by the protonation status of the “isolated” amino acids in solution, which is assumed to be the same when being part of the protein and therefore ignores the amino acids’ local pH environment change due to the complex folding with exclusion of water molecules also around polar amino acids [164].

MD simulation, therefore, provides detailed information on the structure, dynamics, and thermodynamics of proteins and their interactions with other molecules, such as water, ions, ligands, or other proteins. It can also capture conformational changes, folding/unfolding events, and reaction pathways that are difficult or impossible to observe experimentally. However, MD simulation also faces several challenges and limitations, such as the accuracy of the force field, the size and complexity of the system, the length and number of the simulations, and the analysis and interpretation of the results [165].

Force field limitations can be overcome by adopting ab initio methods, i.e., approaches based on the solution of the Schrödinger equation, which is a priori free from parametrization. This approach, nowadays based on the “Density Functional Method, DFT” workhorse paradigm [166], has opened new avenues in the simulation of molecules, polymers, and extended solids. The limitations of DFT are the unfavourable scaling of the needed computer resources with the increasing system size and inaccuracies in treating specific interactions (open shell systems, highly or static correlated systems, etc.). For proteins, due to the large size, DFT seems out of possibilities. Nevertheless, ab initio (at the tight-binding DFT level) protein simulation has been recently reported for the whole HIV-1 capsid in aqueous solution containing more than 62.5 million atoms [167]. Even more demanding has been the simulation of the satellite tobacco mosaic virus with explicit water of solvation [168]. In the geometry optimization, 300 small proteins, an RNA fragment including more than 30,000 atoms, all solvated by explicit water molecules and giving about 1 million atoms, was treated at full DFT level. However, the achievement of these results was only possible by exploiting high-performance computing facilities.

At an even higher level of complexity lies the study of the protein interactions with the surfaces of inorganic materials. To supplement the interpretation of experimental studies, computational methods capable of simulating protein–surface interactions would be an essential complementary approach to the experiments. However, the molecular mechanisms and factors that govern this interaction are still poorly understood, as the standard experimental techniques for pure proteins cannot be applied due to the complexity of the interface (e.g., disorder, solvation effects, molecular motions, etc.). Modelling protein–surface interactions involves several challenges [169]. For instance, while well-developed force fields are available to model both the structure and dynamics of pure proteins (vide supra) and pure oxides [170], much less is known about hybrid force fields which are apt to describe the interface regions between protein and OxPs’ surface sites. On top of that, as explicit water models are usually included in the simulation, it is important not only to describe the water/water and water/protein interactions, but also the water/oxide surface interactions. Despite these difficulties, modelling protein–surface interactions can provide unique insights into the molecular determinants of binding specificity, affinity, orientation, and conformational changes upon binding. For the above challenges, protein–surface modelling is a frontier area of research, deeply influenced by advances in computational hardware, software, algorithms, and data analysis tools. 

In this section, a very selective overview of the protein/surfaces is provided, mainly limiting the focus to a subset of the most recent papers. An extensive review about biomolecules interacting with the silicon oxide surfaces (either crystalline or amorphous) was published a decade ago [76], and in the following section we limit the discussion to silica surfaces as well. A prelaminar step towards protein adsorption is the study of small polypeptides interacting with models of amorphous silica [171]. In that work, a combination of molecular dynamics (MD) simulations and dynamical force spectroscopy experiments based on atomic force microscopy (AFM) were adopted to estimate the free energy of adsorption ΔG_ads_ of a (GCRL) tetrapeptide on amorphous SiO_2_ in pure water. A specific force field was adopted which was capable of dealing with deprotonated silica (mimicking silica in contact with medium at pH > 7) and interacting with water and biomolecules [172]. The authors developed a novel numerical protocol to predict the adhesion work profile along each MD trajectory. The difficulty of these kind of simulations is testified by the large relative error bars of more than 50% on the computed ΔG_ads_ values, which fall within a relatively narrow window between −5 and −9 kcal mol^−1^. 

Conformational changes of a protein due to the contact with inorganic surfaces is a key feature, as the biological activity is highly dependent on the precise conformation of a native protein. This has been analysed for α-chymotrypsin and hen egg white lysozyme adsorbed on amorphous silica using MD force field-based approach in comparison with adsorption experiments [173]. The authors first adopted a very simplified approach to model the protein/surface interaction based on the DLVO theory [174] (named after the initials of Boris Derjaguin and Lev Landau, Evert Verwey and Theodoor Overbeek) in implicit solvent. This approach revealed chymotrypsin pointing the regions of its α -helical toward the surface, due to the large dipole moment, while lysozyme approaches the surface in a side-on orientation. Explicit-solvent MD simulations did not reveal significant conformational changes during the simulation time, while showing the presence of adsorption motifs with both positively/negatively charged and both polar and non-polar residues. Interestingly, stable adsorption originates from a favourable electrostatic complementarity between the adsorption motifs and the local sub-nanometer distribution of charged, strongly hydrophilic, and less hydrophilic surface regions. The important conclusion was that intuitive arguments based on DLVO forces are a good guideline to predict the expected adsorption orientation of the proteins, while fine details in the adsorption features for the two enzymes can only be explained by also considering protein−protein interactions and the explicit atomistic details of the adsorption sites. 

A similar approach was used to follow the conformational changes of chymotrypsin upon adsorption on amorphous silica by comparing predicted versus actual changes in the circular dichroism spectra [175]. The model included the description of the OH groups (Figure 18) with a silanol surface density of 4.4 OH nm^−2^. Calculations showed that chymotrypsin lost part of its helical content upon adsorption, with minor perturbation of its overall tertiary structure. The simulation was able to predict the conformational variations (unfolded/folded/unfolded) in the C-terminal helical fragment in pure water, in solution, and when adsorbed on silica. The theoretical prediction of the circular dichroism spectra from atomistic simulations was critically assessed towards highlighting the limited capability of advanced-sampling MD schemes to explore the conformational phase space of large proteins and the dependency of the predicted ellipticity bands on the choice of calculation parameters.

Lysozyme adsorbed on silica was also the focus of another paper, in which force spectroscopy was targeted by both simulation and experiment [176]. The force spectra provide the force needed to detach the lysozyme from silica surfaces and was recorded by atomic force microscopy (AFM). Ultimately, the measured complex patterns of force peaks did not have a simple interpretation in terms of individual bond-breaking events at the atomic scale. Therefore, the authors provided the atomistic interpretation by means of all-atom steered molecular dynamics (SMD) simulations. In SMD, an external force is applied to drive lysozyme along a given direction, inducing unbinding of the attached groups and conformational changes in lysozyme on time scales accessible to molecular dynamics simulations. 

The required predefined direction for applying external forces may, however, not be necessarily unique for such a complex system. Extra care should be taken in SMD to avoid artifacts due to non-equilibrium effects and irreversible work performed on the system by a force which is too large/fast steering. In that work, the force was applied to the side-chain nitrogen atom of one lysine residue, with a pulling velocity of 5 m s^−1^ in a direction along the silica surface normal which was maintained by strong angular restraints. An essential outcome of this work was the demonstration that the native tertiary structure of lysozyme is preserved if, and only if, its four intramolecular disulphide bridges are intact. Otherwise, the protein pulled off the surface undergoes severe unfolding, which is captured well by SMD simulations in explicit solvent. This is at variance when adopting implicit solvent simulations, which wrongly predicted protein unfolding even in the presence of S–S bridges. This result is an important take-home message: including explicit water in the simulations is essential, as the lack of additional structural stabilization provided by the water’s hydrogen bond network (missing in the implicit water approaches) may give the wrong prediction. In summary, the combined experimental and theoretical results allowed the interpret of the rugged force spectra of lysozyme/silica interfaces, not in terms of successive breaking of internal disulphide bonds leading to partial unfolding events, but rather as the detachment of several molecules bound to the same AFM tip, each anchored to the surface via multiple hydrogen and ionic bonds. 

The contact between biological membranes with inorganic materials either as extended surfaces or as nanoparticles is currently receiving great scholarly interest, and the literature about computer simulation of membranes is vast. Therefore, we only selected one set of review papers to provide a short overview of the field. The first is by Mori et al. [177] which focuses on the enhanced conformational sampling algorithms and explicit/implicit solvent/membrane models for MD simulation. These authors presented several examples of MD simulation studies of membrane proteins, such as glycophorin A, phospholamban, amyloid precursor protein, and mixed lipid bilayers, using different sampling methods and molecular models. Indeed, different generalized-ensemble algorithms, such as replica exchange molecular dynamics (REMD), replica exchange umbrella sampling (REUS), surface-tension REMD, and replica exchange with solute tempering (REST) are described. They also reviewed different implicit solvent/membrane models, such as generalized Born (GB) models, and showed how they can be combined with enhanced sampling methods to improve the efficiency and accuracy of MD simulation. This paper demonstrates how MD simulation can provide valuable insights into the molecular mechanisms of membrane function, while also pointing out some limitations and difficulties of MD simulation for membrane systems, such as those already described for the protein simulation (vide supra). It is therefore not surprising that, due to the increased complexity with respect to the proteins modelling, the atomistic-level simulation between the contact of membranes with inorganic solids is almost absent in the literature, with the work by Heikkila being a notable exception [178]. 

Particularly relevant for the mechanism of silica interactions with erythrocyte membranes is the computational study, at atomistic details, by Delle Piane et al. [179], a dealing with the first steps of the interaction of small silica nanoclusters with the cell membrane (Figure 19). Different silica nanoclusters were modelled in interaction with the membrane of erythrocytes by means of all-atom MD simulations, to predict the free-energy profiles across hybrid interfaces (well-tempered metadynamics). The goal was to elucidate on both the mechanism of silica biomineralization and the reasons behind the haemolytic power of some forms of silica. The considered membranes were a homogeneous bilayer made by POPC, a mixed one in which cholesterol was replacing up to 20% of the POPC lipids and finally, a complex model mimicking the erythrocyte membrane bilayer.

The authors concluded that the predicted free-energy profiles associated with membrane crossing give no evidence for segregation of nanoparticles at the membrane/water interface, irrespective of their Si nuclearity, structure, and charge. However, the associated molecular trajectories, are suggestive of a possible direct translocation mechanism, in which silica nanoclusters elicit both local and large-scale effects on the membrane dynamics and stability. This gives hints towards possible pathways for silica nanotoxicity based on nanoparticle-induced membrane perforation.

Very recently, Pavan et al. [26] employed DFT calculations to model the interactions of PLs (DOPC and DOPS) with silica NFS to rationalize the results obtained by IR spectroscopy and membranolysis tests (see Section 4.2). They concluded that NFS recognise membrane epitopes that exhibit a positive quaternary amino and negative phosphate group. The results demonstrate that NFS content is the primary determinant of membrane disruption causing red blood cell lysis and changes in lipid order in zwitterionic, but not in negatively charged, liposomes.

## 6. Conclusions and Perspectives

There is increasing evidence that the cross-talk between inorganic materials and biomolecules or cells is governed by molecular recognition patterns. OxPs might expose to the outer environment some specific chemical and structural motifs, including well-defined surface OH patterns, whose topochemistry could be “recognized” by structural determinants of a biomolecule. Here, we reviewed experimental, bioanalytical, and modelling techniques to unveil the key OH configurations on OxPs’ surfaces and a new approach methodology (NAM) to investigate the molecular interactions of OxPs with biomolecules and membranes was presented. This NAM was successfully applied to evidence the role of a peculiar surface OH subfamily on silica, i.e., the NFS, and decipher their molecular interaction with choline phospholipids, i.e., the structural determinants of the interaction exposed by cell membranes. This specific interaction was demonstrated to determine the activation of an inflammatory response by lung cells exposed to NFS-rich silica particles and nanoparticles. 

Several complementary physico-chemical techniques are available to determine the acid–base behaviour and electrostatic effects of OH groups in aqueous/biological media (e.g., titration and ζ potential measurements), and to quantitatively determine the surface water and hydroxyl groups content (e.g., thermogravimetry, XPS, NMR spectroscopy). Among other techniques, IR spectroscopy allows the quali-quantitative assessment of several parameters of surface OH groups on particles. The νOH stretching vibration is highly informative about the density, chemical nature, and spatial arrangement of the OH species. Appropriate experimental set-ups, such as outgassing and H–D isotopic exchange with D_2_O vapour, can be used to unveil structural information about the OH groups at the surface of solids, to which the interaction with external biomolecules should be confined. Thermal treatments allow the elimination of all the adsorbed water molecules that could interfere with the assessment of M-OH groups, as well as the selective determination of the different OH families and properties. The use of diffuse reflectance (DR) techniques can help to overcome the issues related to the study of particles with a low specific surface area, which are often of high concern in particle toxicology. 

From a biomolecular point of view, the structural determinants of the interaction with inorganic oxides may consist of specific functional groups or repetitive motifs exposed at the cellular membranes. Among the set of commonly available cells for in vitro studies, erythrocytes are often used as model membranes because of their high predictive power for particle-induced inflammatory activity. The haemolysis assay has been proven to clearly reveal the modifications of the OH-binding properties that could be induced by physical and chemical treatments that altered OH patterns on OxPs. We showed here that the molecular constituents, such as single proteins, phospholipids, or mono/mixed lipidic vesicles, can be successfully used to deconstruct the complexity of the biological membranes. Such an approach represents an adequate strategy to reveal the specific sites of interaction between membranes and OH moieties on inorganic oxides. For instance, a preferential affinity of nearly free silanols for the choline headgroup of PLs has been recently established and proposed as the cause of silica-induced membrane destabilization. 

Unique insights into the molecular determinants of the interactions between membrane/protein and OxPs surface, including binding specificity, affinity, orientation, and conformational changes, could be achieved by computational modelling. The atomistic-level simulation between membranes and inorganic surfaces is, however, still at the early stages. One further step of paramount importance is the inclusion of explicit water in the simulations. The lack of additional structural stabilization provided by water hydrogen-bond network, that is currently not considered in the “implicit water” approaches, may determine inaccurate prediction of binding energies and molecular dynamics. For the above challenges, membrane/protein-surface modelling is a frontier area of research, deeply influenced by advances in computational hardware, software, algorithms, and data analysis tools.

The NAM proposed here was largely applied on SiO_2_ particles to identify determinants of toxicity. Silica is among the most studied OxPs in the biomedical, toxicity, and chemistry fields, because of its widespread natural occurrence, use, and historical occupational health issues. Nonetheless, this approach might be extended to any other hydroxylated surface of inorganic oxides. A number of surface-hydroxylated oxides, including TiO_2_, Al_2_O_3_, graphene oxide, and nanoclays, showed membranolytic effects with variable intensity. However, for most of these oxides, the mechanism of membrane perturbation has not been systematically investigated yet and should be, since they are possibly related to the peculiar topochemistry of some specific surface OH patterns, such as NFS in quartz. The topochemistry of surface hydroxyls can be modulated by material composition and physico-chemical treatments, including surface passivation with organic coatings. These treatments may ultimately expose or hide specific binding sites at the oxide surface and ultimately define the biological fate of the OxPs. 

Overall, the definition of reactive surface OH patterns, as shown for silica, may represent a new paradigm for establishing toxic, catalytic, or more generally, active surface sites that interact with external (bio)molecules. In particle toxicology, a unifying model of particle bio-interaction based on peculiar OH patterns would allow a predictive paradigm to be established for nanoparticle toxicity pathways of metal oxides which are characterized by hydroxylated surfaces. This would be crucial for designing or selecting (nano)materials that exhibit negligible membrane disruption activity.

## Figures and Tables

**Figure 1 ijms-24-11482-f001:**
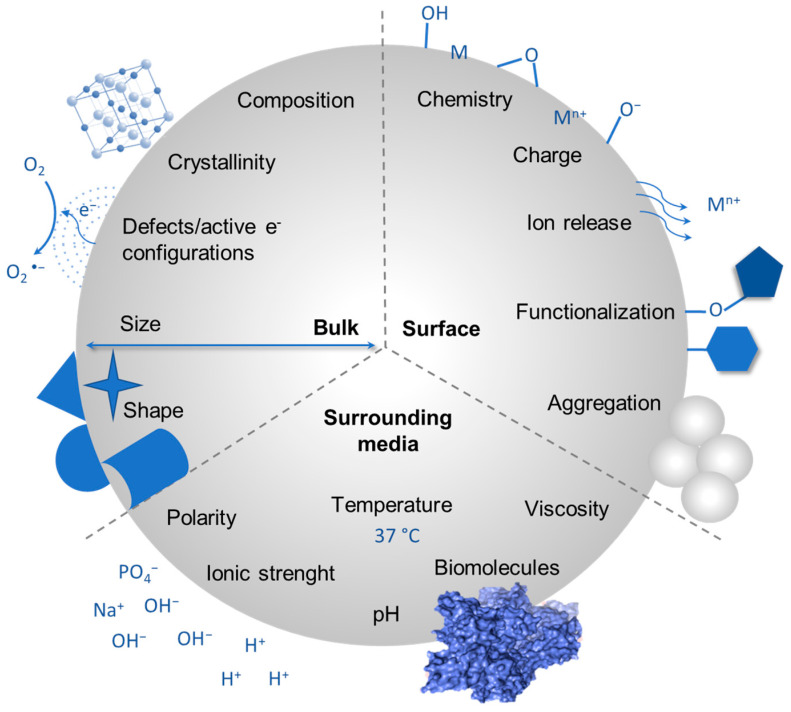
Biophysicochemical factors influencing particle−biomolecule interactions and toxic activity. Several factors including particle composition, electronic structure, surface characteristics, and the biochemical environment, concur to modulate the mechanisms of interaction with biomolecules and membranes, and ultimately the toxic response of the particle.

**Figure 2 ijms-24-11482-f002:**
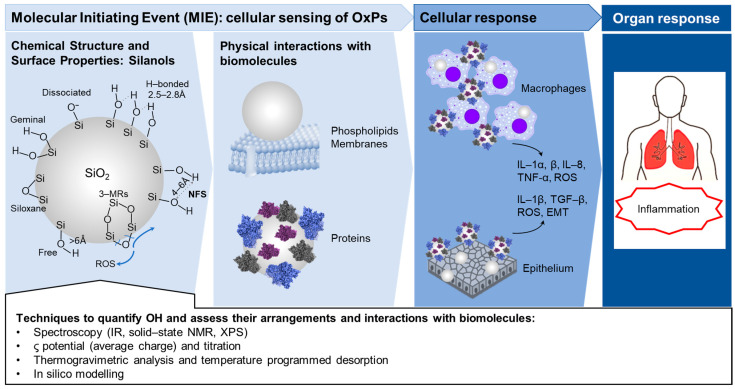
Determination of hydroxyl groups for establishing a molecular initiating event (MIE) in mechanisms of particle toxicity. Examples illustrate possible surface OH (silanols) groups and siloxane patterns for SiO_2_. At a molecular level, these surface features can regulate the interactions with biomolecules and membranes. Biomolecular interactions determine cellular pro-inflammatory and pro-fibrotic responses, leading to organ pathogenic response. Nearly free silanols (NFS) [1] were identified as key determinants of the membranolytic and inflammatory activity of silica.

**Figure 3 ijms-24-11482-f003:**
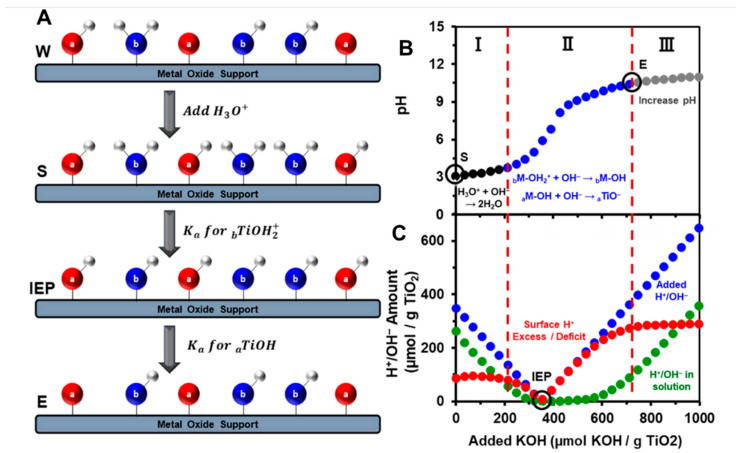
(**A**) Scheme of the acid–base titration method for oxide particles. (**B**) Titration curve for TiO_2_ in water. (**C**) Quantities of H_3_O^+^/OH^−^ added during the experiment (blue) and related concentrations in solution (green) and at the TiO_2_ surface (red). The central region of the graph (region II) allows the key information to be obtained since it contains the isoelectric point (IEP) and the equivalence points for both surface protonated basic hydroxyls (_b_M-OH_2_^+^, in blue) and surface acidic hydroxyls (_a_M-OH, in red). Reprinted with permission from [52]. Copyright 2023 American Chemical Society.

**Figure 4 ijms-24-11482-f004:**
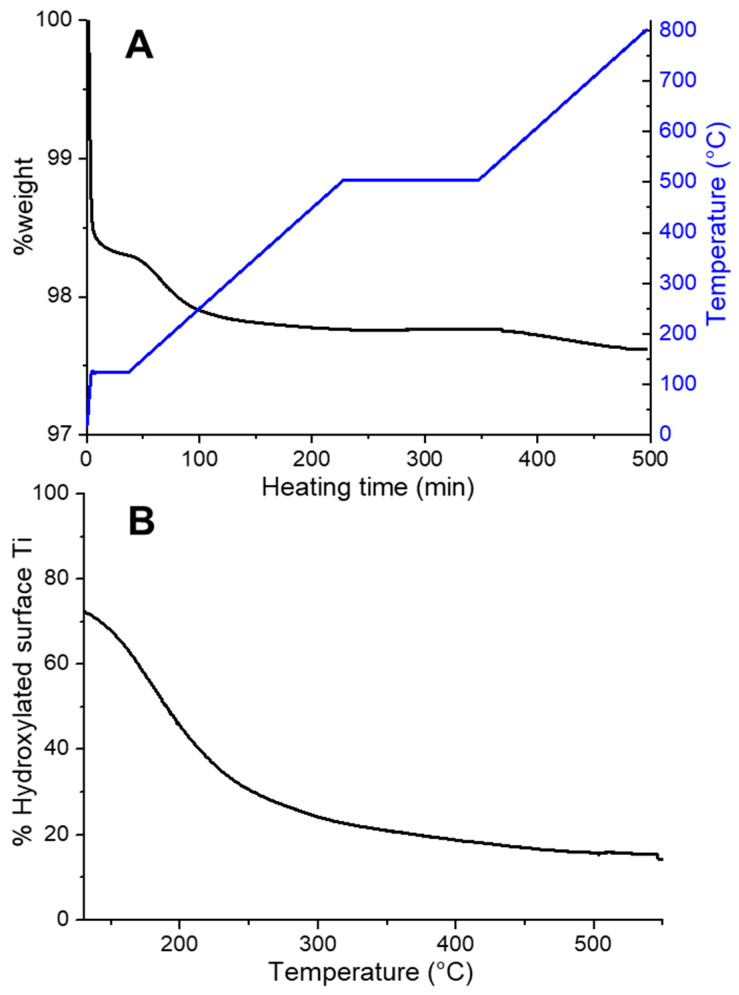
(**A**) TGA curve (black) showing the normalized loss of weight of a TiO_2_ sample as a function of heating time. The corresponding temperature variation (blue curve) as a function of time is also reported in the right axis. (**B**) Percentage of hydroxylated surface Ti^4+^ atoms as a function of the heating temperature calculated from TGA data reported in part (**A**). Adapted with permission from [57]. Copyright 2012 American Chemical Society.

**Figure 6 ijms-24-11482-f006:**
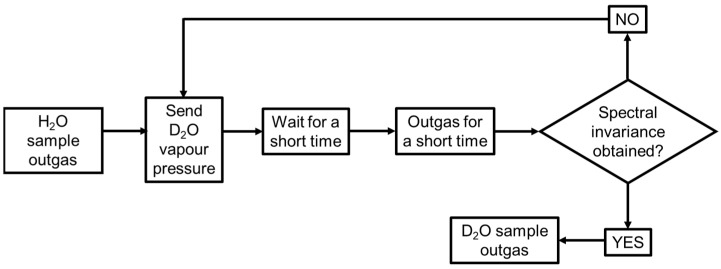
Flow chart that schematically represents the experimental steps to perform an H/D isotopic exchange.

**Figure 7 ijms-24-11482-f007:**
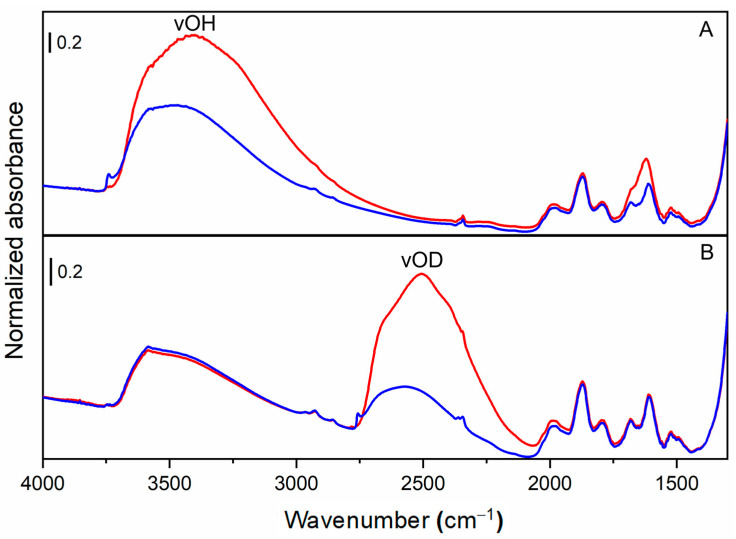
Comparison between the spectra in absorbance acquired before (red) and after (blue) the outgassing procedure for a silica sample at room temperature. (**A**) IR spectra before H/D isotopic exchange. (**B**) IR spectra after H/D isotopic exchange. Adapted with permission from [1].

**Figure 8 ijms-24-11482-f008:**
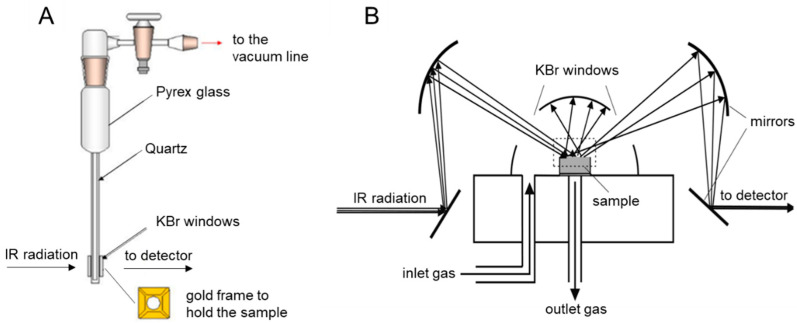
Experimental set-up used in transmission and in reflectance measurements: IR cell equipped with KBr windows (**A**) and scheme of a DRIFT cell (**B**).

**Figure 9 ijms-24-11482-f009:**
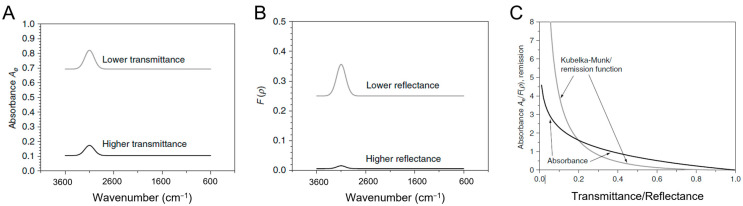
(**A**) Comparison of the different intensity of the IR bands in absorbance depending on the initial transmittance values. (**B**) Comparison of the different intensity of the IR bands in Kubelka–Munk depending on the initial reflectance values. (**C**) Comparison of the absorbance and the Kubelka–Munk functions. Adapted with permission from [88]. Copyright 2009 Elsevier Inc.

**Figure 10 ijms-24-11482-f010:**
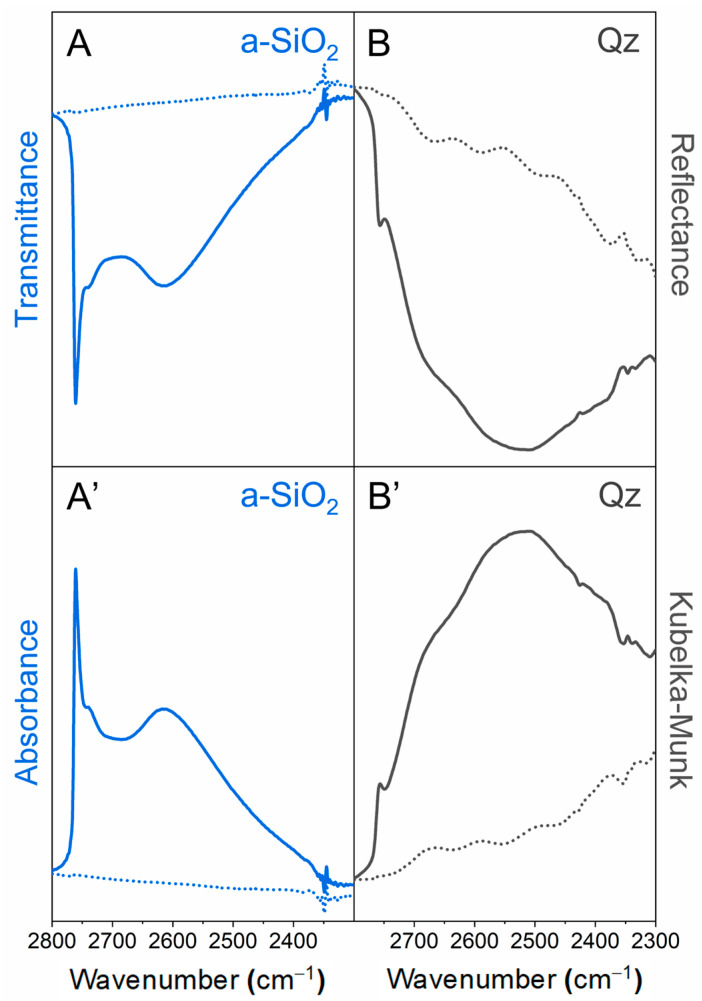
Infrared spectra of an amorphous silica (a-SiO_2_) sample (blue curves) and a quartz (Qz) sample (grey curves) before (dot lines for OH spectra) and after (solid lines for OD spectra) H/D isotopic exchange. The spectra were acquired in transmission mode for a-SiO_2_ (panel (**A**)) and in reflectance mode for Qz (panel (**B**)) and then converted in absorbance (panel (**A’**)) and in K–M function (panel (**B’**)), respectively. The selected range 2800–2300 cm^−1^ shows νOD patterns of surface silanols for the two samples. Adapted with permission from [26,45].

**Figure 11 ijms-24-11482-f011:**
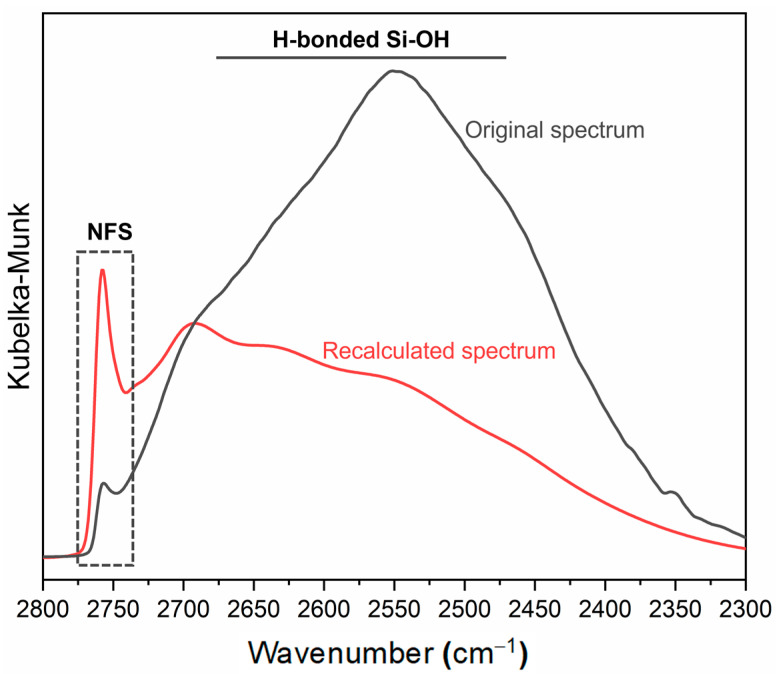
Comparison between the original spectrum (dependent on the specific ε of the different silanol families) and the spectrum recalculated using the Carteret model (dependent only on the relative abundance of the different silanol families) of a quartz sample. The peak at 2758 cm^−1^ assigned to nearly free silanols (NFS) is evidenced in the dotted box. Adapted with permission from [1].

**Figure 12 ijms-24-11482-f012:**
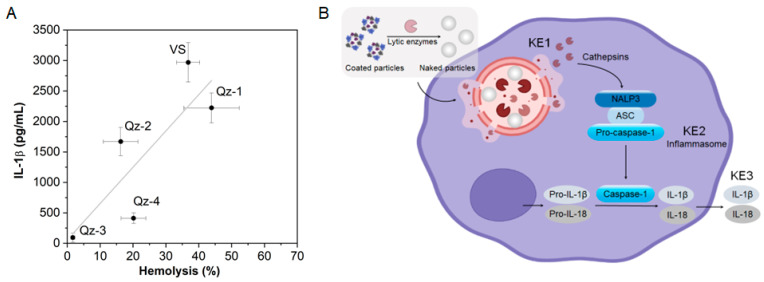
Membranolysis of silica particles is related to their inflammatory activity. (**A**) Haemolysis correlates with IL-1β release from murine macrophages which are challenged with different types of quartz (Qz) and vitreous (VS) silica particles (Person’s linear regression analysis, r: 0.827, r2: 0.683). (**B**) Respirable silica particles in the lungs are internalized by alveolar macrophages (AM) into phagolysosomes. Within the phagolysosome, the particle surface coating due to adsorbed proteins and surfactants of the lung lining layer is removed by lysosomal enzymes. The unmasked silica surface can disrupt the phagolysosomal membrane (key event, KE1). The release of phagolysosomal proteases, such as cathepsins B and S, into cell cytosol triggers the activation of the NALP3 inflammasome machinery (KE2), which leads to activation of the proteolytic enzyme caspase-1 and release of mature pro-inflammatory cytokines (i.e., IL-1β and IL-18) (KE3). This process induces and sustains the inflammatory response in the lungs. Adapted with permission from [96]. Copyright 2014 Springer Nature.

**Figure 14 ijms-24-11482-f014:**
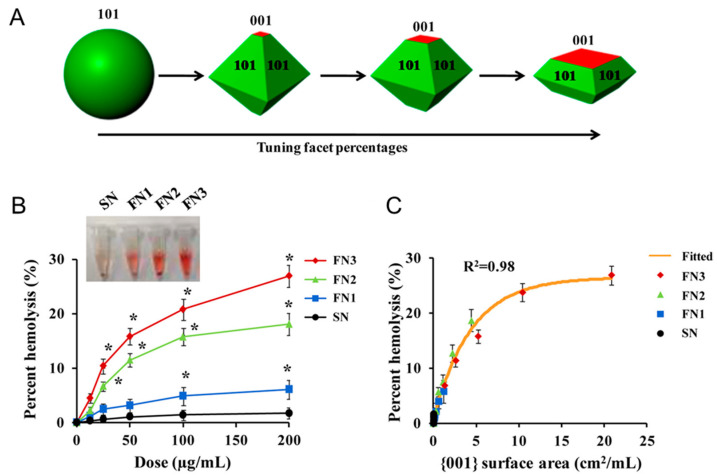
Tuning {001} facet percentage in TiO_2_ nanocrystals correlates with membranolysis. (**A**) Spherical (SN) and truncated octahedral bipyramidal TiO_2_ nanocrystals with increasing {001} facet percentage (FN1–3) were prepared. (**B**) Haemolytic activity of spherical and faceted TiO_2_ nanocrystals at increasing particle concentration. (**C**) Haemolytic activity is correlated to the {001} surface area. Statistical significance was evaluated using two-tailed heteroscedastic Student’s *t* tests with * *p* < 0.05, compared to the untreated cells. Adapted with permission from [113]. Copyright 2016 American Chemical Society.

**Figure 15 ijms-24-11482-f015:**
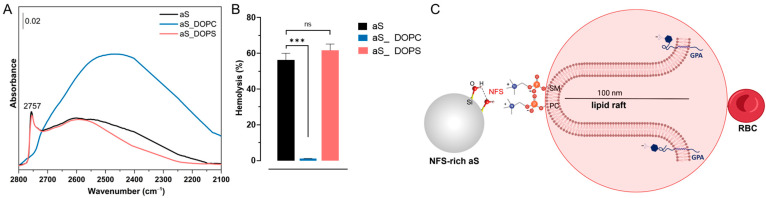
NFS-rich amorphous silica (aS) particles selectively interact with zwitterionic PLs with a phosphocholine. (**A**) Surface silanol distribution (transmittance IR spectra reported in absorbance) in the νOD spectral region (after H/D isotopic exchange and νOH spectra subtraction) of aS incubated with just the vehicle (0.01 M PBS), or 0.1 mg/mL of DOPC or DOPS for 30 min, washed three times with PBS, and dried. (**B**) Haemolytic activity of aS in presence of DOPC or DOPS. Data are the mean ± SEM of three independent experiments and were compared with a two-tailed Student’s *t* test. * ** *p* < 0.001 vs. group without PLs, containing only silica; ns: non-significant. (**C**) Schematic representation of the RBC membrane structure evidencing the major syaloglycoprotein (Glycophorin A, GPA) that confers negative charge to the outer membrane; a sphingomyelin-enriched lipid protrusion deprived of GPA, where sphingomyelin (SM) and phosphatidylcholine (PC) both bearing a phosphocholine are evidenced. Phosphocholine headgroup interacts with NFS on silica. Adapted with permission from R [26].

**Figure 16 ijms-24-11482-f016:**
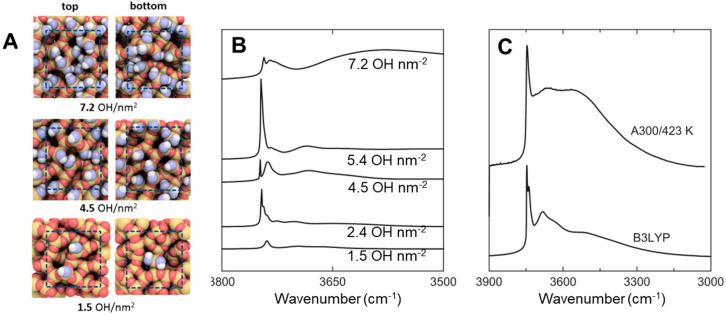
(**A**) Top and bottom views of amorphous silica models with variable OH densities. The oxygen atoms of the surface OH groups are coloured in blue and hydrogen atoms in white. In the bulk structure, Si: yellow, O: red. (**B**) Simulated IR spectra at the DFT B3LYP level for the silica surfaces with increasing OH densities. (**C**) Comparison of an experimental IR spectrum for an Aerosil A300 sample, outgassed at 150 °C, with a simulated IR spectrum, and obtained by a weighted convolution of the separate spectra at different OH densities reported in part B. Adapted with permission from [156]. Copyright 2008 John Wiley & Sons.

**Figure 17 ijms-24-11482-f017:**
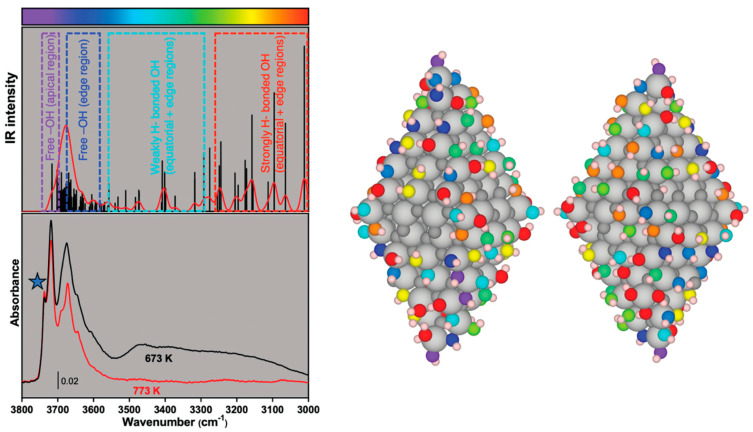
Calculated IR frequencies (**top left panel**) obtained from a realistic OxPs model (**right part**) compared to the IR spectra of TiO_2_ anatase nanoparticles (**bottom left panel**) after outgassing at 500 °C (red curve) or 400 °C (black curve). Vibrational modes are distinguished by a colour scale (from purple to red) which is also used to correlate the OH groups in the model (**right part**) with their respective frequencies. Note that the Ti and O atoms that form the core of the NP are depicted with light and dark grey colours, respectively. The blue star in the bottom left panel indicates the frequency of a single tetrahedrally coordinated OH. Reprinted with permission from [30]. Copyright 2021 Royal Society of Chemistry.

**Figure 18 ijms-24-11482-f018:**
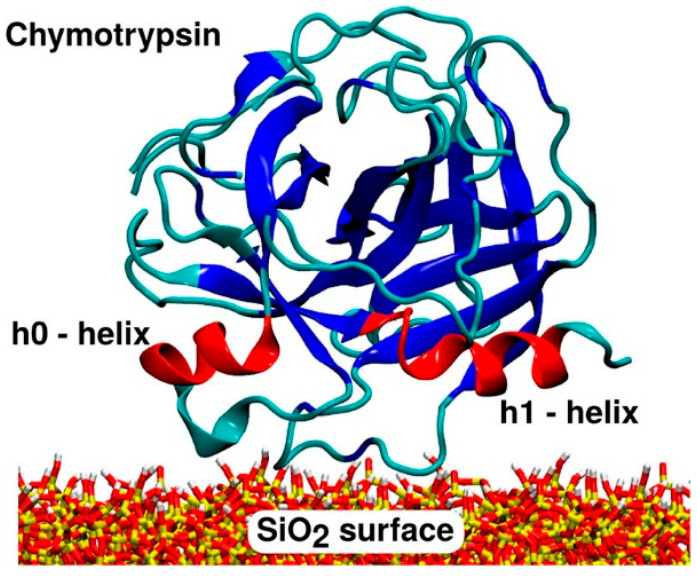
Schematic representation of chymotrypsin in close contact with the silica surface model (Si: yellow; O: red; H: white). The two h0 and h1 helices are highlighted in red. The β-sheets are coloured in blue, whereas the other secondary structure components are visualized in cyan. Reprinted with permission from [175]. Copyright 2018 American Chemical Society.

**Figure 19 ijms-24-11482-f019:**
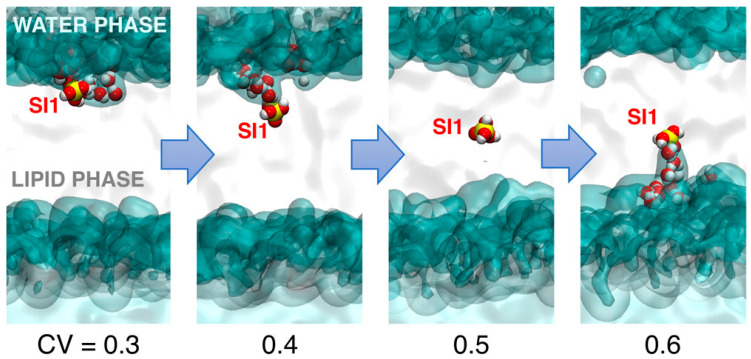
Snapshots displaying the crossing of the POPC lipid bilayer by the SI1 silica cluster. The water and lipid phases are visualized as transparent isosurfaces (cyan and white, respectively), whereas the silica cluster and water molecules close (<5 Å) to it are reported in ball and stick models. The corresponding value of the CV is also provided. Reprinted with permission from [179]. Copyright 2018 American Chemical Society.

## Data Availability

New data supporting reported results, including links to publicly archived datasets analysed or generated during the study, can be requested from the corresponding authors.

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
