# Peer review of "Physico-Chemical Approaches to Investigate Surface Hydroxyls as Determinants of Molecular Initiating Events in Oxide Particle Toxicity"

_ijms, 2023, doi:10.3390/ijms241411482_

Round 1

Reviewer 1 Report

The authors of "Physico-chemical approaches to investigate surface hydroxyls as determinants of molecular initiating events in oxide particle toxicity" wrote an exceptionally well developed and organized review paper that truly captures the state of the art of oxide particle toxicity. This paper was very well written and brought the reader along for a tremendous learning ride. 

There are only a few minor recommendations for this manuscript:
1) Since the title indicates a focus on particle toxicity, I would recommend an expansion on the biological outcomes following particle exposure. While there is a section on this, it is not nearly as comprehensive as the characterization aspects of this manuscript.

2) There are several instances where the paragraphs are exceptionally long, which creates little to no white space on the pages. To improve readability, it is recommended that the long paragraphs be broken up into smaller sections to help readers.

I would recommend another review for minor language edits, but no significant issues were identified. 

Author Response

Reply to Reviewer1’s comments.

The authors of "Physico-chemical approaches to investigate surface hydroxyls as determinants of molecular initiating events in oxide particle toxicity" wrote an exceptionally well developed and organized review paper that truly captures the state of the art of oxide particle toxicity. This paper was very well written and brought the reader along for a tremendous learning ride. 

There are only a few minor recommendations for this manuscript:
1) Since the title indicates a focus on particle toxicity, I would recommend an expansion on the biological outcomes following particle exposure. While there is a section on this, it is not nearly as comprehensive as the characterization aspects of this manuscript.

Response: We gratefully thank the reviewer for her/his positive comment and suggestions for improving the quality of the manuscript.

We agree that a paragraph focusing on the possible toxic outcomes of exposure to OxPs was lacking. We have now reorganized the introduction and added a paragraph that point out the possible toxic outcomes of OxNPs (lines 52 - 69):

“Some tremendous advances in material science and manufacturing are based on innovative use of OxPs. However, the increasing production and use of OxPs result in occupational, environmental, or biomedical exposure scenarios that may generate adverse effects on humans and the living organisms in general. Main routes of human exposure to OxPs are inhalation, ingestion, skin contact, and injection. Inhalation of SiO2 (nano)particles, for instance, is known to induce pulmonary inflammation, fibrosis, and occupational tumour development, especially when workers are exposed to crystalline forms (Pavan et al. 2019; Hoy and Chambers, 2020). Toxic effects of TiO2 particles are re-ported on some types of plants and algae. Moreover, membrane damage to mammalian cells, hepatotoxicity, nephrotoxicity, and pulmonary toxicity have been reported in ro-dents exposed to TiO2 particles (Hou et al., 2019; Ma et al. 2019). Aluminum, cobalt, and iron oxide particles may cause cell membrane damage, cell death, increase in oxidative stress, and genotoxic effects. Studies on ZnO and CuO particles also showed possible cytotoxicity, oxidative stress, and DNA damage (Czyżowska and Barbasz, 2022; Ameh and Sayes, 2019). Toxic effects of a large panel of OxPs were recently reviewed by Sengul and Asmatulu (2020) and El Yamani and coworkers (2022). Through cheminformatics modeling of in vivo, in vitro and in chemico data, both works highlighted the multifactorial parameters that control OxPs toxicity, which can be modulated by size, shape, surface, composition, solubility, aggregation, and particle uptake”.

The present work did not aim to describe in detail the particle toxicity/pathogenicity of OxNPs, on which there are already other comprehensive reviews (see ref. above). We aimed to explore the first molecular initiating event (interaction with membranes and proteins), that could be used as bioanalytical approaches for probing particle surface. We have now better clarified our aim at lines 210-223:

Identifying peculiar reactive OH groups could be mostly useful in defining the Adverse Outcome Pathway (AOP) of OxPs, specifically in establishing the Molecular Initiating Event (MIE), i.e., the interactions between particles and target biomolecules that trigger the early molecular events leading to the pathological outcome (Figure 2). Mechanistically, damage to cell membranes has been reported as one of the main MIE that elicits cellular toxicity, including release of inflammatory and fibrotic mediator and cell death, of many OxPs. In addition to silica, cell membrane disruption has been reported for TiO2, Al2O3, graphene oxide (GO), and many other oxides. Thus, the definition of reactive (in particular, membranolytic) surface OH patterns, as shown for silica (Figure 2), may represent a new paradigm for establishing toxic, but also catalytic, or more in general, active surface sites that interact with external (bio)molecules. In particle toxicology, a unifying model to predict the molecular mechanisms that drive the interactions between particle and biomolecules would allow to propose a predictive paradigm for nanoparticle toxicity pathways of oxides that are characterized by hydroxylated surfaces.

And also at Lines 247-254:

Additional analytical methods to characterize surface hydroxyls and bioanalytical approaches that probe surface reactivity will be also considered to complete the pool of surface techniques aimed at identifying OH groups and defying their interactions with biomolecules.

In this regard, models of cell membranes (i.e., liposomes and red blood cells) and molecular targets of the particle interaction (i.e., phospholipids and proteins) will be used to probe the particle surface reactivity.

2) There are several instances where the paragraphs are exceptionally long, which creates little to no white space on the pages. To improve readability, it is recommended that the long paragraphs be broken up into smaller sections to help readers.

Response: We agree with the reviewer, and we have now reduced the length of several paragraphs to improve readability.

Reviewer 2 Report

The paper is well-written and organized. The references used in this review are ~25% from the last five years (2018-2023), and the number should be increased. The English must be carefully checked (ex. “and is” replaced by “being”). This work can be considered for publication after addressing the following comments, which help the authors to improve their manuscript and the readers through their reading.

Major suggestions:

1.      The para between rows 74-76 has to be updated and correctly written to indicate the role of Zn ions in ROS production, including the molecular mechanisms and targets (thiols, metallothionein, etc.), by direct and indirect ways. There is no obvious direct connection between zinc and ROS, mainly because the bivalent cation zinc does not change its oxidation state in biological systems. Zn ions are linked by their interaction with sulfur, forming the great triad of zinc, ROS, and protein thiols.

2.      A new section about Electron Paramagnetic Resonance (EPR) spectroscopy has to be created with explanations about the surface reactivity of OxPs in dispersions in terms of oxidizing species formation and the intrinsic ability of OxPs to produce oxidizing. The approach with EPR-silent spin trap 1-hydroxy-3-carboxy-2,2,5,5-tetramethylpyrrolidine and monitoring the formation of its derived EPR-detectable radical carboxy-proxyl and the ability of OxPs to magnify oxidizing species formation in an oxidant generating system, i.e., a Fenton-like hydrogen peroxide-catalyzed hydroxyl radical formation detected in the presence of the spin trap 5,5-dimethylpyrroline-N-oxide has to be explained in detail.

3.      All DOI numbers for the cited literature must be provided.

4.      To emphasize the particular features of each technique (method), at the end of the sections, the authors must summarize in a table the NPs (SiO2, TiO2, Al2O3, ZnO, CeO2, and Co3O4, etc.) used, by the techniques, to prove/disprove ROS generation. In the Table, the authors must provide the corresponding references for each method.

5.      Row 576: Please provide permission to adapt Figure 9 from Ref. [74],

6.      Row 768: Please provide permission to adapt Figure 13, “Unpublished Figure reporting data  published in Refs. [99]”

Minor:

1.       Row 67: “Unpublished figure” to be removed

2.       Row 191: “Unpublished figure” to be removed

3.       Row 293: “Unpublished Figure reporting data published in Ref. [43]” to be removed or provide permission from all authors and the journal to use the data (exp Mino).

4.       Row 379: “Unpublished Figure reporting data published in Ref. [1] [44]” to be removed

5.       Row 427: “Unpublished Figure reporting data published in Ref. [1]” -  to be removed

6.       Row 451: “Unpublished Figure” - to be removed

7.       Row 612:   “Unpublished Figure reporting data published in Ref. [14] [33]” -  to be removed

8.       Row 667: “Unpublished Figure reporting data published in Ref. [1]. -  to be removed

9.       Row 938: “Unpublished Figure reporting data published in Ref. [14].” -  to be removed

The English must be carefully checked (ex. “and is” replaced by “being”). 

Author Response

Reply to Reviewer 2’s comments

The paper is well-written and organized. The references used in this review are ~25% from the last five years (2018-2023), and the number should be increased. The English must be carefully checked (ex. “and is” replaced by “being”). This work can be considered for publication after addressing the following comments, which help the authors to improve their manuscript and the readers through their reading.

Response: We thank the reviewer for her/his positive comment, we appreciate the suggestions for improving the manuscript.

The aim of the review was better defined, and the limitations of the work further expanded. We carefully checked the cited literature to increase the most recent references in the last five years and replaced with more recent references, whenever possible. Many methodological references are constrained to the temporal publication period and studies on the toxicity of particles take roots from many well written papers in the early ’80-90s and in the following years.

Major suggestions:

  1. The para between rows 74-76 has to be updated and correctly written to indicate the role of Zn ions in ROS production, including the molecular mechanisms and targets (thiols, metallothionein, etc.), by direct and indirect ways. There is no obvious directconnection between zinc and ROS, mainly because the bivalent cation zinc does not change its oxidation state in biological systems. Zn ions are linked by their interaction with sulfur, forming the great triad of zinc, ROS, and protein thiols.

Response: We have revised this paragraph according to the suggestion provided by the reviewer: “Cu2+ and Zn2+ ions leached out from oxides such as CuO and ZnO are well-known inducers of oxidative stress and pulmonary inflammation (Zhang et al. 2012). Zn2+ ions cannot directly participate in redox reactions, but these can interact with sulfur, forming the triad of Zn, ROS, and protein thiols, such as in the case of metallothionein (Hübner and Haase, 2021). Similarly to CuO, Fe3O4 and particles that are contaminated by ferrous ions (Fe2+) are known to catalyse the Fenton or Haber–Weiss reactions and generate OH radicals in solution [8].”

  1. A new section about Electron Paramagnetic Resonance (EPR) spectroscopy has to be created with explanations about the surface reactivity of OxPs in dispersions in terms of oxidizing species formation and the intrinsic ability of OxPs to produce oxidizing. The approach with EPR-silent spin trap 1-hydroxy-3-carboxy-2,2,5,5-tetramethylpyrrolidine and monitoring the formation of its derived EPR-detectable radical carboxy-proxyl and the ability of OxPs to magnify oxidizing species formation in an oxidant generating system, i.e., a Fenton-like hydrogen peroxide-catalyzed hydroxyl radical formation detected in the presence of the spin trap 5,5-dimethylpyrroline-N-oxide has to be explained in detail.

Response: According to the reviewer’s suggestion, we have better clarified the limits of our work and the focus on methods to probe surface OH features, which can be read as an alternative SAR to the capacity of particles to generate oxidative in the mechanism of toxicity of OxPs:

Lines 133-135 “Hence, besides the intrinsic capacity of OxPs to generate oxidative stress, additional surface features should be involved in defining the mechanisms of OxP harmful interactions with biological systems and the environment, with the potential to generate toxicity.”

Lines 242-262 “In this review, we present a new approach methodology (NAM) based on IR spectroscopy for identifying the surface OH sites that are responsible for the early events taking place when a xenobiotic particle enters a living organism, i.e., the interactions with biomolecules. The large-scale application of this approach to investigate the hydroxylated surface of oxides will favour the discovery and the consolidation of MIE in (nano)particle AOPs. Additional analytical methods to characterize surface hydroxyls and bioanalytical approaches that probe surface reactivity will be also considered to complete the pool of surface techniques aimed at identifying OH groups and defying their interactions with biomolecules.

In this regard, models of cell membranes (i.e., liposomes and red blood cells) and molecular targets of the particle interaction (i.e., phospholipids and proteins) will be used to probe the particle surface reactivity. Experimental interaction models can be validated at the molecular scale by computational approaches. Since many of these aspects have been already clarified for silica, most of the examples will be drawn from studies related to silica, but the possible extension of this NAM to other OxPs will be also discussed. Far from being a comprehensive description of all the currently reported molecular mechanisms in OxPs toxicity, this work is limited to illustrating the physico-chemical and bioanalytical approaches that can be used to reveal and describe surface OH of OxPs, mostly in a toxicological perspective. Several other features have been proposed to be involved in the toxicity mechanism of OxPs, such as the intrinsic capacity to generate ROS, and they have been extensively reviewed elsewhere [48].”

We understand the Reviewer’s interest for a paragraph that describes EPR spectroscopy. A following comment (#4) also suggests including in the work a table for proving/disproving the role of ROS generation in the toxic outcomes of OxPs. These two requests, which have an intrinsic value and could be exploited in future work, are largely beyond the scope of the present work and would further extend what we believe is a quite exhaustive piece of work on the role of OH moieties and the most suitable methodologies available to analyze them, in the context of oxide toxicity. Incidentally, the role of ROS in the mechanism of toxicity of OxPs and the methods to measure these species have been already considered in previous reviews (for instance, Horie and Tabei 2021 10.1080/10715762.2020.1859108; Ayres et al. 2008 10.1080/08958370701665517; Øvrevik et al. 2015 10.3390/biom5031399; Li et al. 2008 10.1016/j.freeradbiomed.2008.01.028; Xia et al. 2006 10.1016/j.coem.2006.07.005). To follow the reviewer’s suggestion and to increase the exhaustiveness of our manuscript, we have added those references to the text.

  1. All DOI numbers for the cited literature must be provided.

Response: We have now provided the DOI numbers for all the cited literature.

  1. To emphasize the particular features of each technique (method), at the end of the sections, the authors must summarize in a table the NPs (SiO2, TiO2, Al2O3, ZnO, CeO2, and Co3O4, etc.) used, by the techniques, to prove/disprove ROS generation. In the Table, the authors must provide the corresponding references for each method.

Response: Please consider response to point 2.

  1. Row 576: Please provide permission to adapt Figure 9 from Ref. [74],

Response: We have now provided the required permission.

  1. Row 768: Please provide permission to adapt Figure 13, “Unpublished Figure reporting data published in Refs. [1, 15, 99]”

Response: We have removed the statement since the figure has been completely modified with respect to that published before.

Minor:

  1. Row 67: “Unpublished figure” to be removed
  2. Row 191: “Unpublished figure” to be removed
  3. Row 293: “Unpublished Figure reporting data published in Ref. [43]” to be removed or provide permission from all authors and the journal to use the data (exp Mino).
  4. Row 379: “Unpublished Figure reporting data published in Ref. [1] [44]” to be removed
  5. Row 427: “Unpublished Figure reporting data published in Ref. [1]” -  to be removed
  6. Row 451: “Unpublished Figure” - to be removed
  7. Row 612:   “Unpublished Figure reporting data published in Ref. [14] [33]” -  to be removed
  8. Row 667: “Unpublished Figure reporting data published in Ref. [1]. -  to be removed
  9. Row 938: “Unpublished Figure reporting data published in Ref. [14].” -  to be removed

Response: We have removed the statement “Unpublished figure” from all the mentioned figures. We have replaced the statement for Figure 4 and provided the required permission.

Comments on the Quality of English Language

The English must be carefully checked (ex. “and is” replaced by “being”). 

The text has been checked by a person proficient in written scientific English and the suggested correction has been made to the entire text.

Round 2

Reviewer 2 Report

Thank you very much to the authors for improving the manuscript. Even new section and explanation about Electron Paramagnetic Resonance (EPR) spectroscopy have not been created, the manuscript can be accepted for publication in the present form.